# Reducing the number of load cases for fatigue damage assessment of offshore wind turbine support structures by a simple severity-based sampling method

Lars Einar S. Stieng[1] and Michael Muskulus[1]

[1]Department of Civil and Environmental Engineering, Norwegian University of Science and Technology NTNU, Trondheim, Norway

**Correspondence:** Lars Einar S. Stieng (lars.stieng@ntnu.no)

**Abstract.** The large amount of computational effort required for a full fatigue assessment of offshore wind turbine support structures under operational conditions can make these analyses prohibitive. Especially for applications like design optimization, where the analysis would have to be repeated for each iteration of the process. To combat this issue, we present a simple procedure for reducing the number of load cases required for an accurate fatigue assessment. After training on one full fatigue analysis of a base design, the method can be applied to establish a deterministic, reduced sampling set to be used for a family of related designs. The method is based on sorting the load cases by their severity, measured as the product of fatigue damage and probability of occurrence, and then calculating the relative error resulting from using only the most severe load cases to estimate the total fatigue damage. By assuming this error to be approximately constant, one can then estimate the fatigue damage of other designs using just these load cases. The method yields a maximum error of about 6% when using around 30 load cases (out of 3647) and, for most cases, errors of less than 1-2% can be expected for sample sizes in the range 15-60. One of the main points in favor of the method is its simplicity when compared to more advanced sampling-based approaches. Though there are possibilites for further improvements, the presented version of the method can be used without further modifications and is especially useful for design optimization and preliminary design. We end the paper by noting some possibilities for future work that extend or improve upon the method.

## 1 Introduction

The large number of environmental states that need to be considered for design of offshore wind turbine support structures is a significant challenge. A simulation is required for each such state, often referred to as a load case, when analyzing the response of these structures to the offshore environment. Each simulation of this kind, at least when carried out with accurate aero-elastic software, is a non-trivial task in terms of computational effort. Assessing the structural performance in the fatigue limit states for operational conditions alone typically means thousands of load cases when following relevant standards (International Electrotechnical Commission, 2009). Consequently, the computational effort needed in total presents a challenge. The increasing availability of high performance computing clusters in both the industry and at academic institutions has alleviated this issue somewhat for one-time assessments of single designs, but there are other contexts where the problem remains relevant. Design

optimization (Muskulus and Schafhirt, 2014; Chew et al., 2016; Oest et al., 2017) in particular is such a case, where having to do repeated structural analyses of evolving designs means that the inclusion of thousands of load cases becomes highly prohibitive. Hence, there is a need for methods that can reduce the computational effort of these analyses, preferably without losing too much accuracy. Motivated by this need, the present study concerns itself with the development of a method that

reduces the number of load cases that need to be analyzed down to a more manageable level. Though other loading scenarios are in general relevant, the present work will focus on sets of load cases encompassing the fatigue assessment of operational conditions for the wind turbine.

Several previous studies in the area of simplifying fatigue assessment through load case reduction have been carried out. Zwick and Muskulus (2016) looked at two different methods, piece-wise linear approximation and multi-linear regression,

to simplify fatigue analysis for a jacket subject to 21 operational load cases. Using varying wind speeds, with a lumped sea state, the approach aimed to train the methods using fatigue data from several jacket designs and then to use them to predict the fatigue damage of other designs. With this approach, the authors obtained reduced load case sets with sizes of 3-6, with maximum prediction errors for the total fatigue damage of about 6% when using 3 load cases. One limitation with this study was that extensive training of the methods, with substantial computational effort, was required in order to obtain these results. The

number of load cases studied was also small compared to the complete set of operational conditions. Häfele et al. (2017) and Häfele et al. (2018) used an approach where reduced load case sets were derived by sampling distributions for the probability of occurrence of the various environmental states, taken from a database of 2048 states. From a hierarchy of load case subsets, the authors estimated the fatigue damage for several different jacket designs. Though the errors were quite high for the smallest subset sizes, this approach demonstrated a clear potential for large reductions in computational effort. Velarde and Bachynski

(2017) used a fatigue design parameter in order to select only the most important sea states for detailed fatigue assessment of a monopile.

Multiple studies of load case reduction have also been conducted for floating support structures. Müller et al. (2017) formulated an approach that combined a response surface model with Latin Hypercube Sampling and an artificial neural network. Müller and Cheng (2018) studied an approach making use of Sobol sequences in order to select the optimal load cases to

sample. This lead to a more rapid convergence than would have resulted from using just conventional Monte Carlo methods. The approach achieved a maximum error of about 10% in the fatigue estimates when using reduced load case sets of 200-500 out of a total of 5400. Finally, Kim et al. (2018) used an artificial neural network to modify the stress transfer function in order to simplify fatigue assessment in the frequency domain.

While achieving various degrees of success in terms of accuracy and ability to reduce the computational effort, a common

trait in most of the cited studies above are that their aims differ slightly from ours. These studies, the one by Zwick and Muskulus (2016) exempted, tried to simplify the fatigue assessments of single designs by making use of methods that were based on considerations of the environmental states alone. Whereas we aim to use also information about the actual fatigue damage for each load case of a base design and then use the combined information to develop a reduced sample set that can be used for designs that have been altered compared to this base design. Since the latter approach is highly relevant for applications

like design optimization, we think the present study addresses a gap in the literature.

The method proposed in this study, like in many of the cited studies above, is based on the idea that there is a large amount of information about the total fatigue damage contained in a small subset of the load cases. Furthermore, a fundamental assumption for this method is that the relative fatigue response to each load case remains approximately constant for an extended family of related support structure designs. This makes it possible to train the method on one full fatigue analysis, using the complete set of load cases, and then use the method to propose which load cases should be assessed for future analyses of designs that have been modified. The method itself is based on sorting the load cases by their contribution to the total fatigue damage and then obtaining the partial sum of their contributions, up to a certain, smaller number of load cases. The relative difference between this partial sum and the total fatigue damage is assumed to be constant when the underlying support structure design is modified. From the corresponding partial sum of any new design, multiplied by a scale factor derived from the original relative difference, the total fatigue damage of that design can then be obtained. Hence, using an approach relying simply on sorting and summation, an estimate for the total fatigue damage based on a significantly reduced set of load cases is readily available.

## 2   Background and methodology

Even when restricting the area of study to operational loading conditions and fatigue analysis for the support structure, there is a substantial amount of work that has to be carried out in order to verify that the structure satisfies design requirements. Keeping in accordance with the standards means covering a lot of different environmental conditions (International Electrotechnical Commission, 2009) and following a specific procedure for calculating the fatigue damage (Det Norske Veritas, 2016). Every realization of wind and wave conditions corresponds to a single load case $E$, which has a probability of occurrence $P(E)$. After a time domain analysis of the support structure, subject to the loading conditions encoded by $E$, the time series of normal stress is estimated in eight different points along the circumference of each relevant location in the structure. The fatigue damage can be found from the stress by performing rainflow counting (Rychlik, 1987), applying SN-curves (DNV GL, 2016) for each stress range identified and then accumulating the damage by the Palmgren-Miner rule. The maximum fatigue damage value found among the eight points along the circumference of a given location in the structure is chosen to represent the fatigue damage per load case, $D(E)$, of that specific location. The total fatigue damage from all load cases, $D_{\text{tot}}$, during a lifetime $T_{\text{lt}}$, at a specific location in the structure, is then given by:

$$D_{\text{tot}} = T_{\text{lt}} \cdot \sum_E P(E)D(E) \tag{1}$$

A central fact to note here is that the contribution of each load case $E$ to the total fatigue damage is determined by the product of the individual fatigue damage and the probability of occurrence. So the most *severe* load cases in the sense of having the largest contribution to the sum are in fact those where there is a balance between these two factors. Very small damage and high probability, or vice versa, tend to give smaller contributions. Whereas load cases incurring intermediate fatigue damage while also having reasonably high probability of occurrence, tend to be the most severe. This will be important below in determining which load cases get sampled. Normally, a safety factor would be applied to Eq. (1). However, since this only changes the

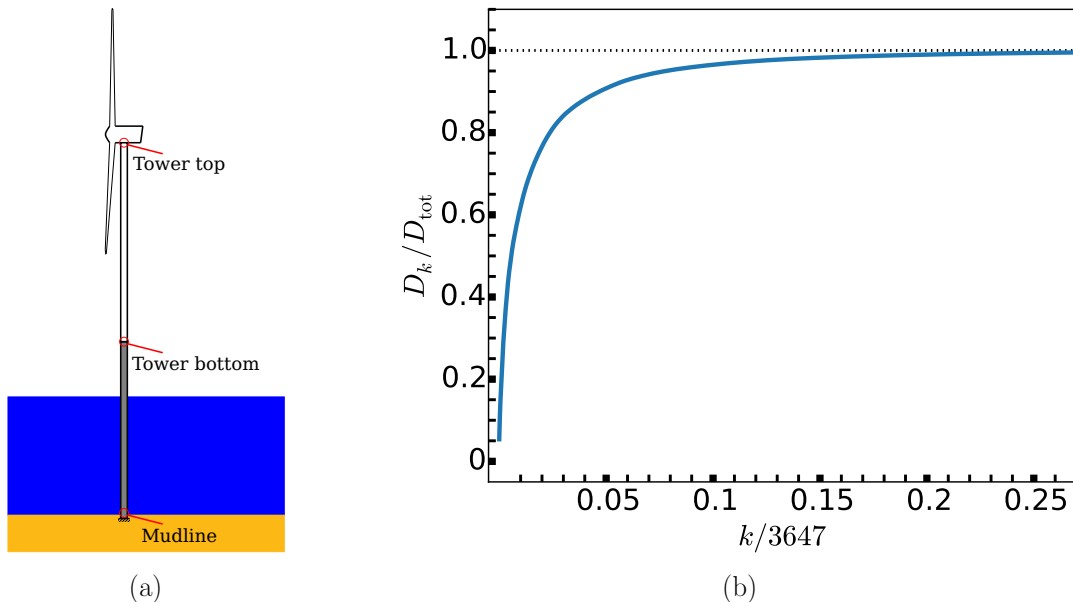

**Figure 1.** Illustration of the model used in this study (a) and a plot of the curve formed by the partial sums, $D_k$, as a function of the number of load cases used, $k$, after sorting the load cases by their severity (b).

result by a fixed constant, it has been neglected here. By the same reasoning, the lifetime scale factor $T_{lt}$ will also be neglected from now on.

### 2.1 Sampling based on the $k$ most severe load cases

From Eq. (1), we can define the $k$-th partial sums of the fatigue damage as:

$$D_k = \sum_{i=1}^{k} P(E_i)D(E_i) \tag{2}$$

If we now let the set $\{E_i\}$ of load cases be sorted in descending order based on the size of the corresponding product of probability of occurrence and fatigue damage (from now on called *severity*), then from experience $D_k$ should start to get close to $D_{tot}$ after values of $k$ corresponding to only a few percent of the total number of load cases. In fact, plotting these partial sums as a function of $k$ gives a curve like the one shown in Fig. 1(b), from which the previous statement can be confirmed. This curve was calculated using data from the tower bottom, but the corresponding curves at other locations in the structure show exactly the same behavior. Furthermore, we may define the relative difference between the sorted $k$-th partial sum and the actual total fatigue damage as:

$$\epsilon_k = 1 - \frac{D_{tot}}{D_k} \tag{3}$$

As our fundamental approximation, we may assume that $\epsilon_k$ is constant when the underlying support structure design is modified. That is, suppose we want to estimate the total fatigue damage $D_{\text{tot}}^{\text{new}}$ of some new designs of the same basic support structure, with corresponding $k$-th partial sums $D_k^{\text{new}}$. If we assume that $\epsilon_k = \epsilon_k^{\text{new}}$, then we can obtain an estimate for the new total damage as:

$$\hat{D}_{\text{tot}}^{\text{new}} = D_k^{\text{new}} - D_k^{\text{new}} \cdot \epsilon_k \tag{4}$$

The intuitive interpretation here is essentially that the new total damage is the $k$-th partial sum plus (since $\epsilon_k$ is always negative) an error term that should make up the difference. Some further clarity can be obtained by simplifying the above:

$$\hat{D}_{\text{tot}}^{\text{new}} = D_k^{\text{new}} \cdot (1 - \epsilon_k)$$
$$= D_k^{\text{new}} \cdot \frac{D_{\text{tot}}}{D_k}$$
$$\hat{D}_{\text{tot}}^{\text{new}} = D_{\text{tot}} \cdot \frac{D_k^{\text{new}}}{D_k} \tag{5}$$

Hence, in practice, the estimate for the new total fatigue damage is the old total fatigue damage times the ratio of the new $k$-th partial sum to the old $k$-th partial sum.

## 2.2 Sampling for multiple locations

If we only wanted to know the total fatigue damage at a single location in the structure, Eq. (5) would suffice. However, there is a slight complication when the fatigue damage at multiple locations is needed. While for the most part we expect the order of the severity of the load cases to be about the same at every location, there is no guarantee that it will be *exactly* the same. Hence, using information from just a single location to decide which load cases to sample could lead to significant errors at the other locations. The simplest solution to this is to take the union of the most severe load case sets from each location. Specifically, let $V_k^i$ be the set of the $k$ most severe products $P(E)D(E)$ at location $i$. We can then define the sampling set, $\tilde{V}_k$, as:

$$\tilde{V}_k = \bigcup_i V_k^i \tag{6}$$

Specifically, we combine the $k$ most severe load cases from each location into an expanded set (removing any duplicates), from which we then calculate the partial sums to be used in Eq. (5). It would also be possible to define the sampling set in such a way that it would have an already given size, filling up with as many load cases from the individual location sets as possible. Motivated by, for example, having certain restrictions on how many load cases one can afford to sample given the computational resources and the task at hand. However, this would result in an unbalanced set, biased towards one or more of the locations. Hence, it would be preferable to let the sizes of the individual sets determine the size of the sampling set and then simply choose a value of $k$ such that the resulting sampling set size is acceptable.

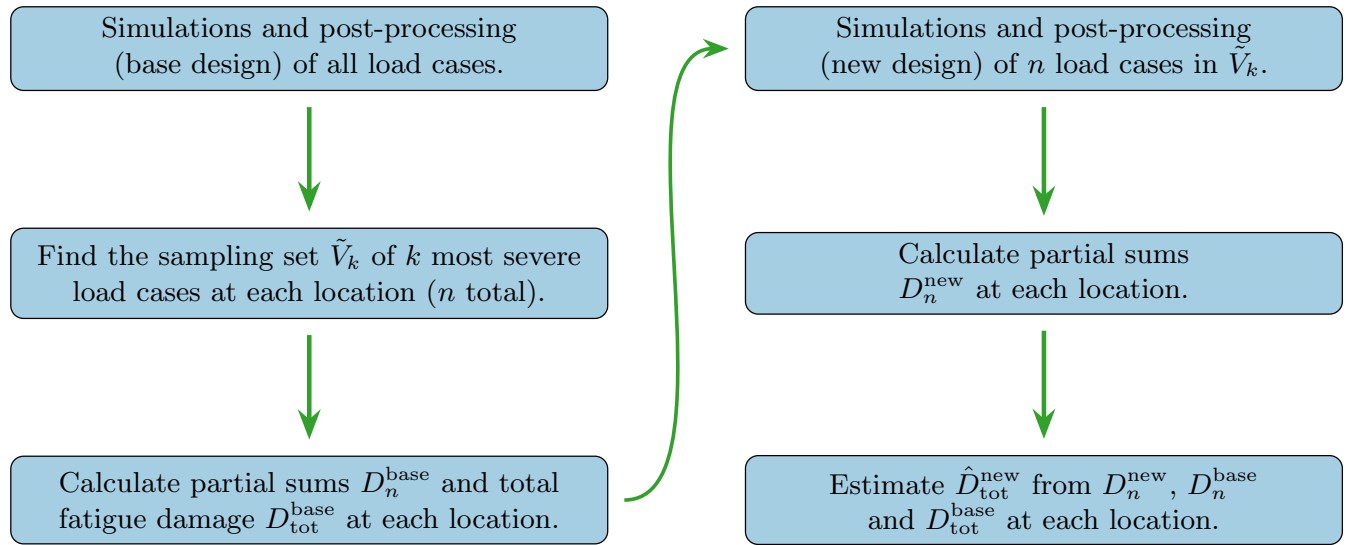

**Figure 2.** Step by step summary of the estimation method.

### 2.3 Fatigue damage estimation procedure

By using one full fatigue assessment of a base design, we can then train our method on this data. Sorting the load cases by the severity at each location and then taking the union of the resulting sets, we obtain the sampling set $\tilde{V}_k$ for a given number of $k$ load cases from each location. If we denote the size of the sampling set by $n$, the $n$-th partial sums at each location $i$ of the base

5    design, $D_n^{\text{base},i}$, combined with the corresponding total fatigue damage, $D_{\text{tot}}^{\text{base},i}$, are then used to define $\epsilon_n^i$. The total fatigue damage estimate at location $i$ for any new design, $\hat{D}_{\text{tot}}^{\text{new},i}$, is then obtained by performing simulations and fatigue assessments for the $n$ load cases in the sampling set, estimating the new $n$-th partial sums as

$$D_n^{\text{new},i} = \sum_{j=1}^{n} \tilde{V}_{k,j}^{\text{new}} \tag{7}$$

where $\tilde{V}_{k,j}^{\text{new}}$ is the set of $n$ severity values obtained for the new design, and finally scaling the base total fatigue damage as

10    prescribed in Eq. (5):

$$\hat{D}_{\text{tot}}^{\text{new},i} = D_{\text{tot}}^{\text{base},i} \cdot \frac{D_n^{\text{new},i}}{D_n^{\text{base},i}} \tag{8}$$

The procedure is summarized in Fig. 2.

### 2.4 Simulation setup and testing framework

For the simulations used in this study we have used the fully integrated aero-elastic software tool FEDEM Windpower (Fedem

15    Technology, 2016). Our model is comprised of the NREL 5MW turbine (Jonkman et al., 2009) sitting atop the OC3 monopile support structure (Jonkman and Musial, 2010). The structural model was built using three-dimensional Euler-Bernoulli beam

elements connected by nodes, one at each end of the elements. At each node there are six degrees of freedom and the internal forces and moments can be automatically estimated and exported for further post-processing. The monopile model was clamped at the seabed. The external wind loads were estimated from turbulent wind field time series given as input. The wave loads were calculated within the software itself by explicit generation of waves from a JONSWAP spectrum according to specified

wave parameters and using the Morison equation with drag and added mass coefficients equal to 1.0. Marine growth was included in the model, but current was not. The load cases used in the study have been derived from the Ijmuiden Shallow Water Site wind and wave data reported by Fischer et al. (2010), giving probabilities of occurrence for different wind speeds, sea state parameters and wind and wave misalignment. The selected environmental states represent wind speeds between 4 and 24 m/s with bin sizes of 2 m/s (giving 11 different speeds) with a given turbulence intensity for each wind speed, significant

wave height and peak period values depending on wind speed (between 21 and 42 different realizations for each speed) and incoming wave directions varying between 0° and 330° in steps of 30° (giving a total of 12 directions for each sea state and wind speed). 3647 load cases were used in total. One simulation of length 10 minutes (after removing initial transient data) was used for each load case, including different random seeds for each realization of wind and wave input. To make the study more tractable, only one random seed per 10 minute simulation was used, rather than the 6 seeds (or alternately using a single

60 minute simulation) usually required by standards. However, the reason for this requirement is that the fatigue damage per load case becomes more stable, i.e. less subject to statistical fluctuations, when additional seeds (or simulation time) are added. Hence, if the method can be shown to work for fatigue damage values calculated based on a single 10 minute simulation per load case, the method would certainly also work when using 6 random seeds or more. In order to test the method, three different locations along the height of the support structure, thought to be representative of different response behaviors, were selected.

These include the tower top, the tower bottom and the mudline. A drawing of the model, which includes identification of the selected locations, is shown in Fig. 1.

As noted previously, one of the main motivations for this study has been applications to design optimization. Hence, we have found it pertinent to test our method in a setting that would resemble situations likely to be encountered during an optimization loop. Starting with an initial support structure design on which the method is trained, how well would the method perform

in predicting the fatigue damage of the modified designs encountered during the optimization? In other words, we want to see how the method performs on designs that correspond to configurations that might represent intermediate steps, or even something close to a solution, of a design optimization problem. This prompts a few different strategies for how to obtain these modified designs. First of all, the type of optimization framework we want to investigate here is mass (or weight) optimization. In this framework, the diameters and thicknesses of various elements are changed until the design is as light as possible,

while satisfying certain constraints on structural performance. To see how the method would perform during an optimization procedure of this type, we chose designs where the element diameters and thicknesses had been scaled either up or down compared to an original design. To represent different types of scenarios, the scaling was done both systematically across the entire structure and randomly from element to element. For each of these strategies and for two different magnitudes of scaling, the elements of the structure were scaled once according to the given strategy and a new design was thus obtained. In total,

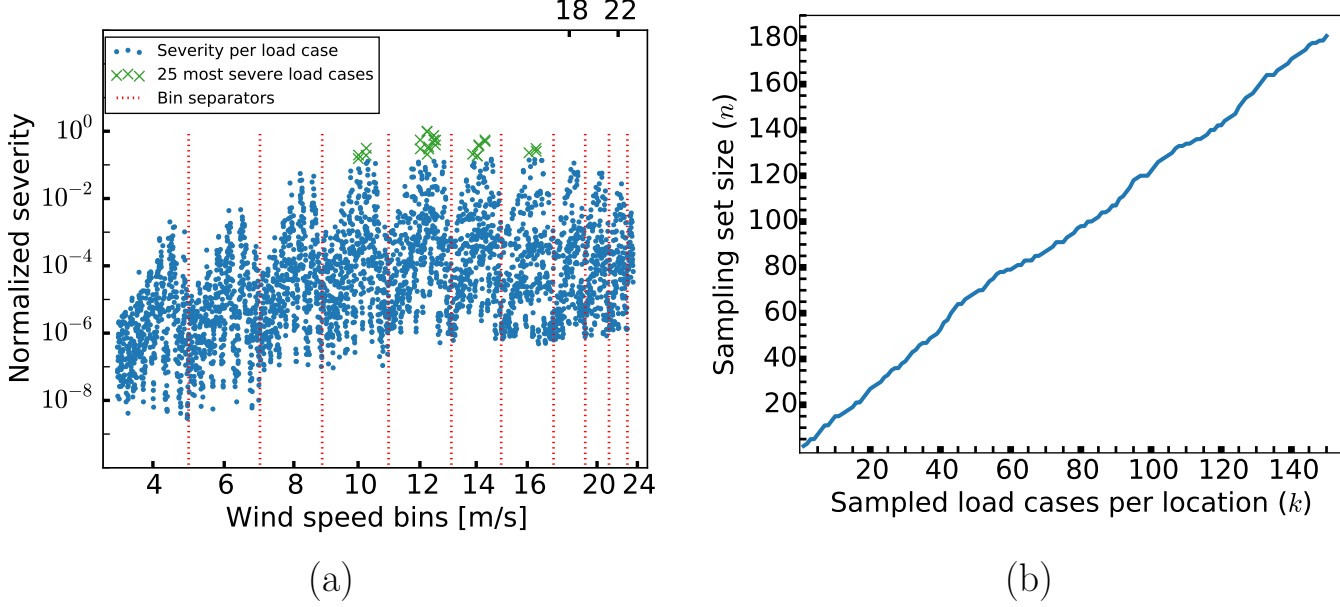

**Figure 3.** Normalized severity per load case at tower bottom, with load cases separated into different wind speed bins, with the 25 most severe load cases specially marked (a) and the size of the total sampling set as a function of the number of load cases included from each of the three locations (b).

seven new designs were generated. Their names (for easy reference later) and quick summaries of how each design was scaled is given in Table 1.

## 3 Results

As an initial point of entry, we may ask which of the load cases are in fact the most severe for the base design and hence 5 which ones will be sampled by the method. From the distribution shown in Fig. 3(a), it is clear that the most severe load

**Table 1.** The modified designs used in this study, with names and how they have been modified (scaled).

| Design name | Design description |
|---|---|
| MD5 | Element sizes scaled down by 5% |
| MI5 | Element sizes scaled up by 5% |
| MR5 | Element sizes randomly scaled up or down by 5% |
| MD10 | Element sizes scaled up by 10% |
| MI10 | Element sizes scaled down by 10% |
| MR10 | Element sizes randomly scaled up or down by 10% |
| MRU10 | Element sizes randomly scaled up or down by up to 10%, using a uniform probability distribution |

cases are clustered among just a few wind speeds. In particular, these speeds are (in order of which speed has the highest number of severe load cases) 12 m/s, 14 m/s, 16 m/s and 10 m/s. Though less clear from the plot, these load cases otherwise represent the wind and wave misalignment angles and sea state parameter values having the highest probability of occurrence. In other words, while the severity of the wind speeds is a result of a balance between incurred fatigue damage and probability

of occurrence (at the particular site used in this study, 6 m/s has the highest probability of occurrence among the wind speeds), the severity of particular wind and wave misalignment angles and sea state parameter values within a given wind speed bin is dominated by the probability of occurrence. The analysis here is based on data taken from the tower bottom, but completely analogous conclusions can be drawn from the two other locations.

For each support structure design listed in Table 1 and the unaltered base design, a full fatigue analysis was performed

(that is, not just for the load cases selected by the method) in order to be able to quantify the performance of the method. Specifically, the performance of the method has been quantified in a way similar to Eq. (3), using now the relative difference, $\delta$, of the estimate and the true value for the total fatigue damage of each design:

$$\delta = 1 - \frac{\hat{D}^{\text{new}}_{\text{tot}}}{D^{\text{new}}_{\text{tot}}} \tag{9}$$

One concern might be that there are large differences in the order of the severity for the load cases in each of the three locations.

This would in principle lead to sampling sets that are very large compared to the number of load cases selected per location. However, our results indicate that this is not the case. A plot of the size of the total sampling set as a function of the number of load cases selected from each location is shown in Fig. 3(b). It is reasonably linear, varying between $n = 7$ for $k = 5$ and $n = 181$ for $k = 150$. Hence, as an approximation, $k$ can be said to be fairly close to the actual number of sampled load cases, at least for smaller sample sizes. Finally, for the sake of not showing data that yield little additional insight, the results shown

below are in each case taken from a single location only. Specifically, for each design, the location with the maximum error was chosen to represent the behavior of all three locations within a given plot. In practice, the chosen location is usually either the tower bottom or the mudline, since the behavior at tower top seems generally more favorable.

### 3.1 Uniformly scaled designs

In Fig. 4, the relative errors, $\delta$, for various sampling set sizes, $n$, is shown for the four uniformly scaled designs (MD5, MI5,

MD10, MI10). Except for in the case of MI10, the estimates fairly quickly converge to a level of roughly 2% error or less. For MD5 and MI5 this level of accuracy requires 20-30 samples (a reduction in the load case set by more than a factor 100), whereas for MD10 it takes about 50 samples to reach this level (though at 30 samples the error is no more than 3%). For MI10, the convergence is slower and the error is generally a bit higher. In this case, the error level is at around 6% or less after 30 samples, goes below 5% at around 100 samples and then slowly tends toward 4% or less for the larger samples sizes. The

maximum error encountered is at about 13% for MI10 and is otherwise less than than 10% for the other designs. In other words, for the the first three designs, errors of about 4-8.5% are attainable using only 7 load cases. We observe that the method seems to consistently over-predict the fatigue damage (giving negative errors, see Eq. (9)) when the design has been consistently scaled down and under-predicts (giving positive errors) when the design has been scaled up. Inspecting Eq. (5) we may surmise that

this means that for down-scaled designs the proportion of the fatigue damage in the $k$-th partial sum has increased compared to the base design, whereas for the up-scaled designs this proportion has decreased. The overall convergence is not quite smooth, presenting some occasional jumps in the estimation error. These jumps are ultimately quite small (usually at no more than a single percentage point) and are likely signs of small instabilities in the method for reduced sample sizes. In these cases, the sudden inclusion of certain additional load cases (with the effect of either improving or decreasing performance) can have a visible effect on the overall estimate. As for why MI10 seems to under-perform when compared to the others, this is likely because the changes to the global eigenfrequency induced by scaling all elements by 10% can lead to dynamic amplification for lower wind speed load cases when the frequency increases (corresponding to the structure being scaled up). In this particular situation, there is a significant shift towards the 3P frequency of the turbine, defined as the rotation speed-dependent frequency at which any of the three blades passes by a fixed point, as seen from the Campbell diagram of the NREL 5MW turbine (Jonkman and Jonkman, 2016). The result is a significant increase in the severity of lower wind speed load cases, which means that the error in including only the most severe load cases in the fatigue estimation changes more drastically for this design. This in turn makes the method less accurate than for the other designs, where the changes in fatigue damage are more uniformly distributed among the load cases.

In Fig. 5 the same results are shown for some selected larger sets of load cases, including also some smaller sets of load cases for reference. The accuracy of the method keeps increasing as the number of load cases used increases, but the gain in accuracy for each additional load case (as indicated by the slope of the curve) decreases drastically after a certain point. After 183 load cases, corresponding to a reduction of the number of load cases by a factor 20 and corresponding to the smallest error shown in Fig. 4, the benefit is fairly minor. While a further reduction in error by one order of magnitude can be achieved by the use of 730 or 911 load cases, the error at 183 load cases is already small enough that the cost in additional computational effort is likely prohibitive. The exception is again for model MI10, where the convergence is much slower and the errors generally higher. Going from 183 load cases to 730 (a reduction factor of 5 compared to the full set of load cases) takes the error from around 4% to around 1.5%.

## 3.2 Randomly scaled designs

The relative errors, $\delta$, in the estimates for the randomly modified designs (MR5, MR10 and MRU10) are shown in Fig. 6. These all generally show improved performance compared to the uniformly modified designs. Except for the smallest sample estimate for each model, every estimate has an error of less than 2%. For MR5 and MRU10, errors of no more than 1% occur with sample sizes of no more than 35-40 (a reduction of the load case set by a factor of about 100). MR10 crosses this same error threshold at around 50-60 samples. There is in general a reasonable convergence behavior for all three models. MR10 exhibits marginally higher errors than the two other models. This could be because element scaling of $\pm 10\%$ could lead to a higher degree of overall uniform changes than in the other cases. Since each element in the structure has a different size, one would expect a certain bias towards either overall decrease or increase when the scaling is done randomly from a uniform distribution. The larger the scaling, the larger the resulting bias. In fact, inspecting the changes to the overall mass for these models, MR10 has a bias twice as large as MR5. However, this bias does not consistently lead to over- or under-prediction of

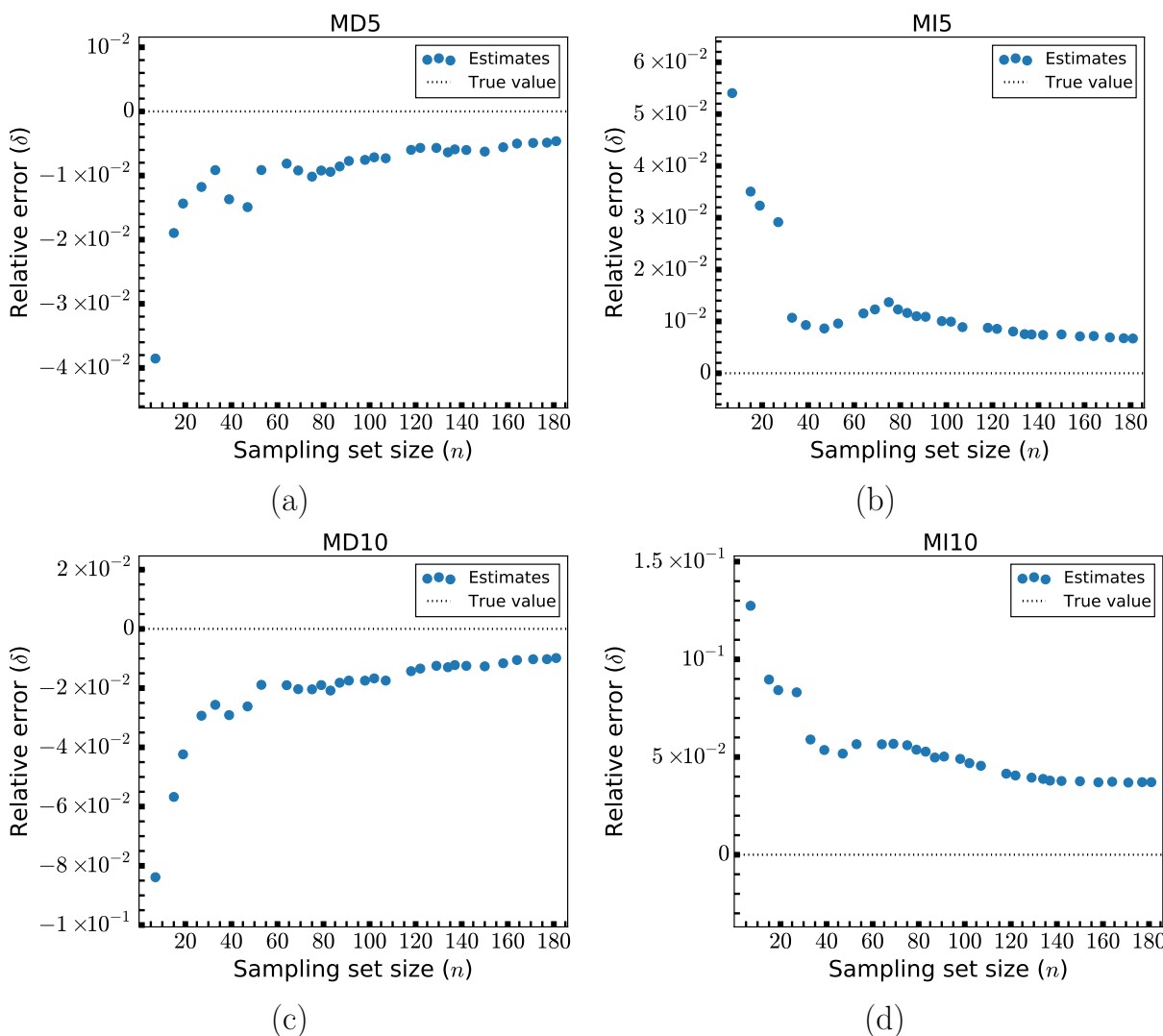

**Figure 4.** Relative errors, $\delta$, of fatigue estimates for models MD5 (a), MI5 (b), MD10 (c) and MI10 (d).

the fatigue damage like it did for the uniformly scaled models. While the results shown here makes it look like the behavior is the same as before, this is not the case for all the points in the structure.

In Fig. 7 results for larger number of load cases for the randomized models are shown. As was similarly noted for Fig. 5, the benefit of increasing the number of load cases beyond what was shown in Fig. 6 is small when considering the speed of convergence of the error and its overall order of magnitude at that point.

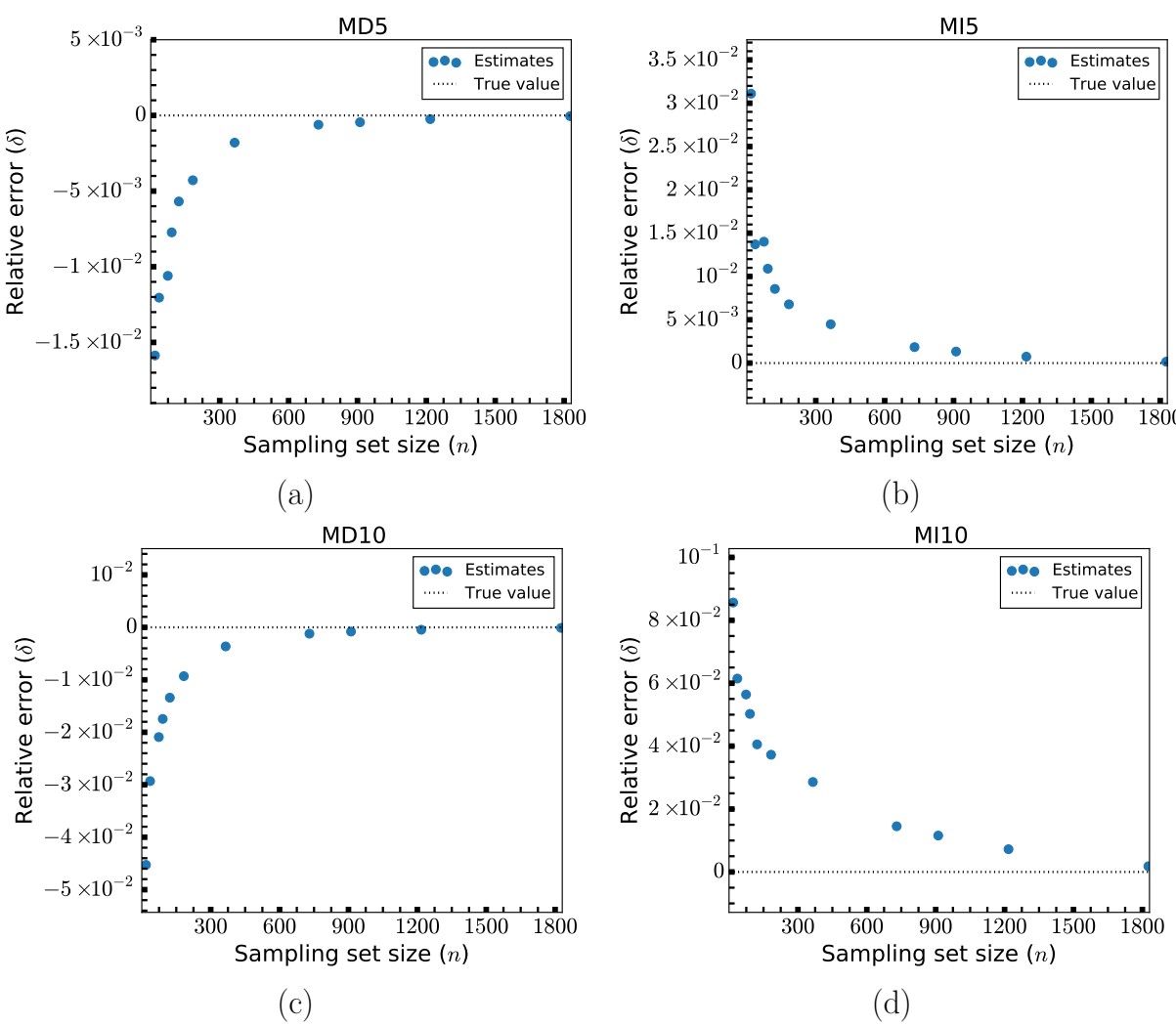

**Figure 5.** Relative errors, $\delta$, of fatigue estimates for larger sets of load cases; models MD5 (a), MI5 (b), MD10 (c) and MI10 (d).

### 3.3 Real behavior of $\epsilon_k$

When we initially defined the method, it was based on a basic assumption: That the relative error when using only the $k$ most severe load cases would remain approximately constant under modification of the support structure design. The results shown so far indicate that this is indeed the case, but this should be verified explicitly in order to have confidence in the theoretical basis of the utilized methodology. To investigate this, we have calculated the absolute value of the relative difference between the value of $\epsilon_k$ (as defined in Eq. (3)) for the base design, $\epsilon_k^{\text{base}}$, and the actual value of $\epsilon_k$ for each modified design, $\epsilon_k^{\text{new}}$. This is shown, together with the respective values of $\epsilon_k$ as heatmaps in Fig. 8, where the color of each cell indicates the absolute value of the relative difference. There is generally very good agreement between the values of $\epsilon_k$ for each design, though again there

is some more deviation for design MI10. This is presumably for reasons similar to why the estimation method had larger errors in this case. The attentive reader might notice that unlike above, the relative differences do not decrease for increasing values of $k$. If anything they seems to either fluctuate or increase. However, as can be seen by inspecting the numerical values, the absolute differences do decrease. Hence, while the relative differences increase, the actual numerical values of these differences

become quite small and therefore less relevant in practice. Furthermore, it is not hard to show from Eq. (3) and Eq. (5) that:

$$\left| 1 - \frac{\epsilon_k^{\text{new}}}{\epsilon_k^{\text{base}}} \right| = D_{\text{tot}}^{\text{base}} \cdot \frac{\left| 1 - \frac{D_{\text{tot}}^{\text{new}}}{\hat{D}_{\text{tot}}^{\text{new}}} \right|}{D_{\text{tot}}^{\text{base}} - D_k^{\text{base}}} \tag{10}$$

Where $D_{\text{tot}}$ refers to the real total fatigue damage, $D_k$ is the $k$-th partial sum of severity products as defined in Eq. (2), the ˆ refers to an estimate made using the method introduced in this study and the superscripts "base" and "new" refer to the initial design and any modification of this design respectively, as above. Both the numerator and the denominator tend to zero as

$k$ tends to the total number of load cases, so the actual behavior depends on the convergence of the method (controlling the numerator) compared to the proportion of the total fatigue damage in a given partial sum $D_k$ (controlling the denominator). Essentially, one can roughly compare the convergence shown in Fig. 5 and Fig. 7 to that shown in Fig. 1(b). In other words, since the denominator converges faster than the numerator, the relative difference in Eq. (10) will tend to increase for increasing values of $k$. A practical consequence of this, which was also noted previously, is that the benefit of increasing $k$, i.e. including

more load cases in the estimate, becomes very small after a certain point. Additionally, since the convergence of the fatigue estimates for MI10 was particularly slow, the behavior seen in the heatmaps for this design at both tower bottom and mudline (a significant increase for increasing $k$) seems reasonable.

## 4    Further discussion

### 4.1    Viability of the method

As seen above, the proposed method is able to predict the total fatigue damage of the modified designs with a high degree of accuracy. With the exception of design MI10, all estimates eventually converge towards errors of 2% or less (in some cases much less) and with drastic reductions in the load case set (factors of 50-200 in most cases). Even for the case of MI10, where the error is about 4-6% for all but the smallest sample sizes, this result is quite convincing in terms of the level of accuracy that can be expected for such an approach given the extent of the modifications to the structural models. In fact, higher accuracy

than that reported for design MI10 might not even be required. A 5% error in the prediction of total fatigue damage represents a change in the lifetime of a support structure by 1 year if the real expected lifetime is 20 years. This is certainly within the range of other types of errors one might expect in terms of uncertainties in the modeling or the environmental conditions, both of which are usually accounted for by multiplying the total fatigue damage by partial safety factors of 2-3. In such a framework, errors on the order of 10% might even be acceptable, in which case a very large load case reduction is possible for

all models. Additionally, there seems to be a clear connection between consistent changes to the size (mass) of the structural elements and whether the estimates for the fatigue over- or under-predict the true value. In fact, the two properties are directly

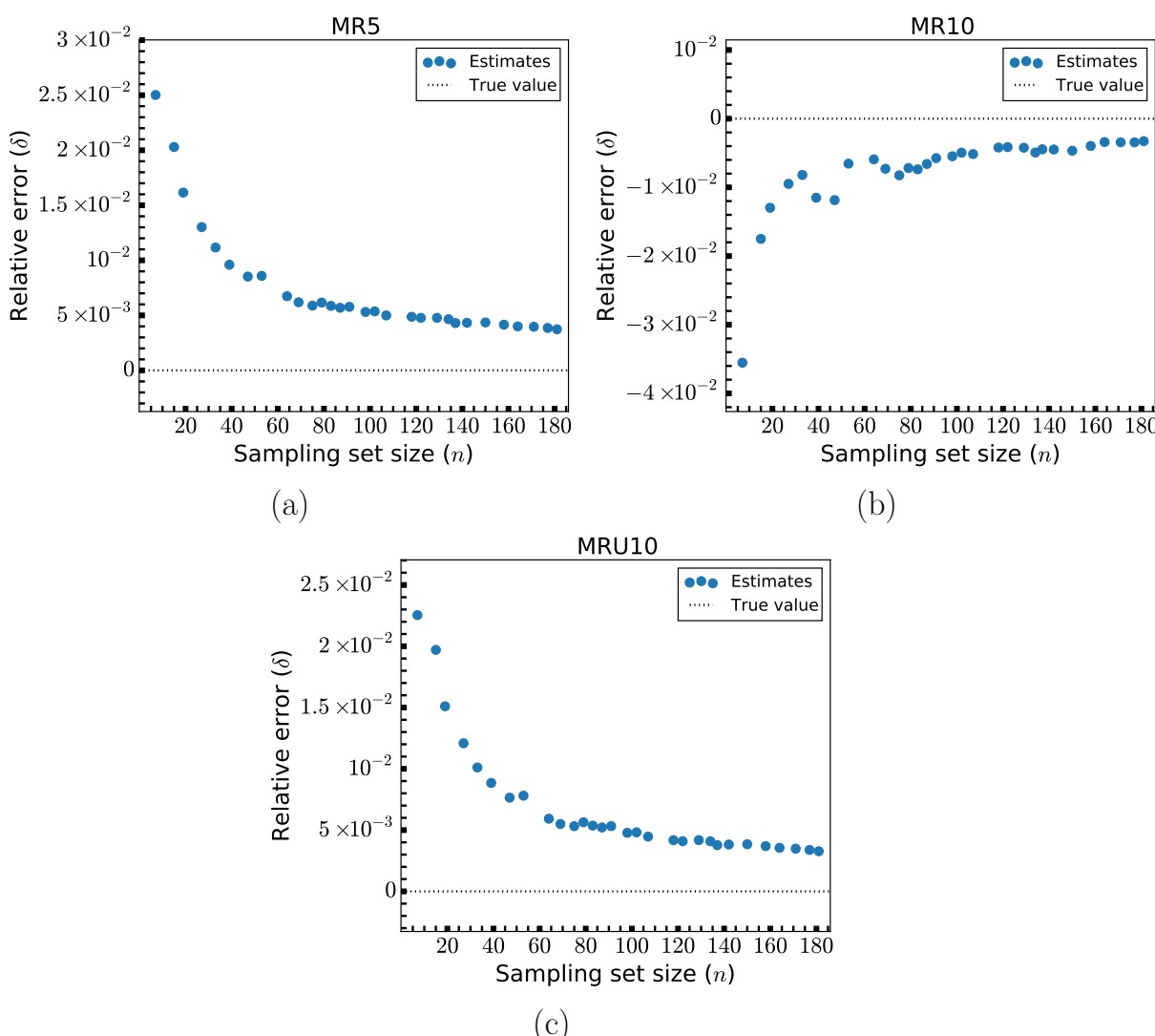

**Figure 6.** Relative errors, $\delta$, of fatigue estimates for models MR5 (a), MR10 (b) and MRU10 (d).

correlated. Though in practice the consequence of under-predicted fatigue damage is much more severe than over-predicted fatigue damage, the fact that the correlation is as visible as indicated in Fig. 4 and Fig. 5 means that it is possible to correct for this effect. Keeping track of systematic changes in the structure hence makes it possible to account for the errors in the estimates in correspondingly systematic ways. For instance, if the estimate is known to be an over-prediction, then it may be deemed "safe" in a conservative sense and in the opposite case one might want to add in a small safety factor. However, the results for the randomly modified designs indicate that overall changes to the structural mass is not enough to account for this behavior. Correspondence between the overall changes and the changes to the elements where the fatigue damage is calculated

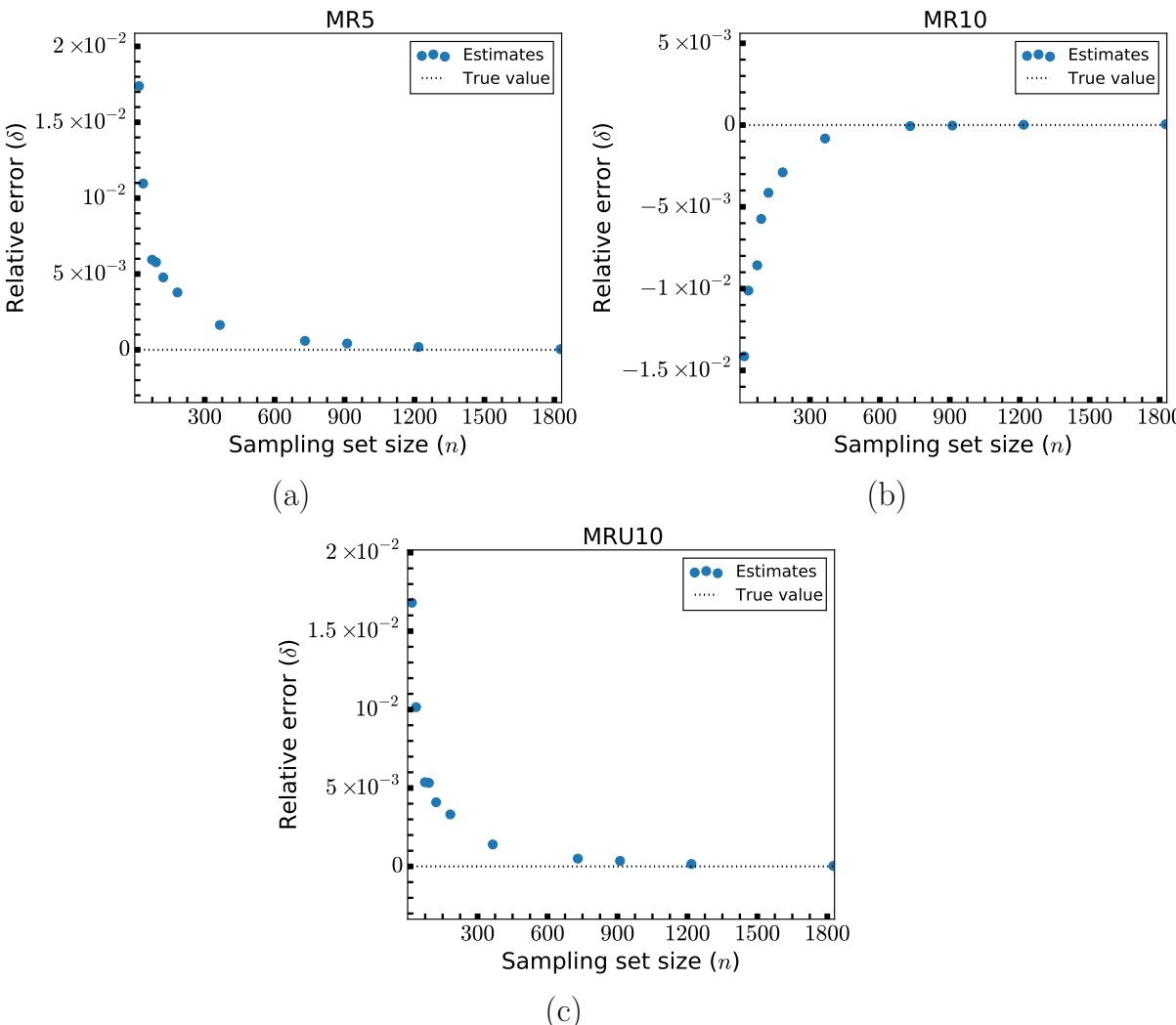

**Figure 7.** Relative errors, $\delta$, of fatigue estimates for larger sets of load cases; models MR5 (a), MR10 (b) and MRU10 (d).

is also important. Hence, some care must be taken when attempting to correct for consistent over- or under-prediction of the fatigue damage.

One of the reasons the method is as efficient as it is when analyzing more than one location in the structure, is the behavior seen in Fig. 3(b): That the number of load cases in the sampling set does not increase significantly when considering all three locations. Since the three locations chosen are so far away from each other, located at each end of the support structure and around the middle of the structure respectively, we do not expect that the addition of even more locations should make a significant difference. However, we do note that in the worst case scenario where there is no overlap between the set of the $k$ most severe load cases at each of the $l$ locations with that of any of the other locations, the size of the sampling set would

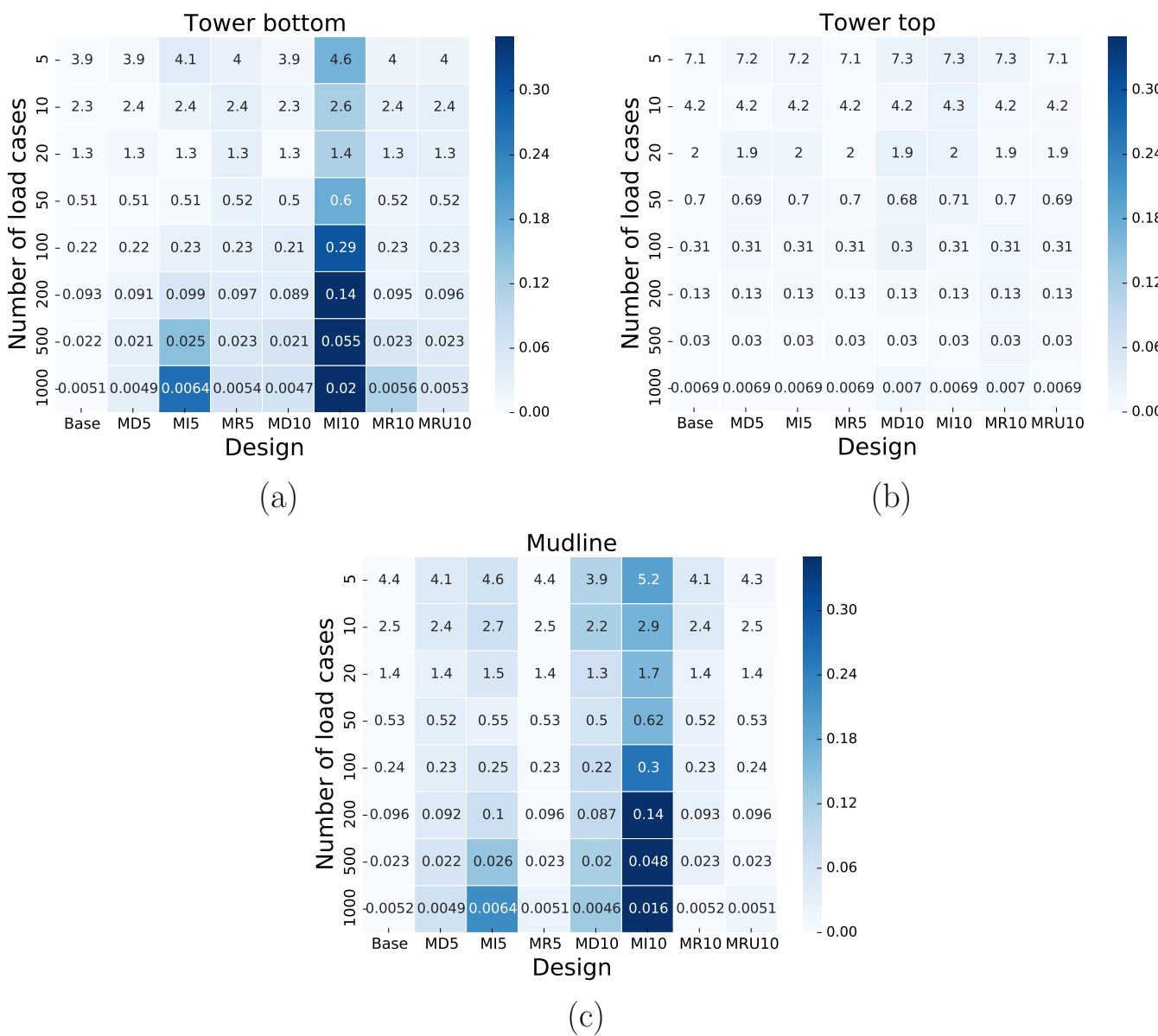

**Figure 8.** Values of $\epsilon_k$ for the base design and each additional design, for selected values of $k$. The colors of each cell are set by the magnitude of the relative difference between the base design value and the value in the given cell. At tower bottom (a), tower top (b) and mudline (c).

be $k \cdot l$, drastically reducing the efficiency of the method. Though our results indicate that anything close to this behavior is unlikely, at least for any monopile support structure, some attention should be paid to ensure similar performance if applying the method to other types of support structures.

The methodology has been shown to be quite effective for a range of different support structure designs, but there is one limitation that should be noted: The presented results were all obtained while using the same turbine model. The turbine model will have a very important impact on global dynamics, e.g. 1P and 3P frequencies and the total system mass, damping and stiffness, and it is hence likely that changing turbines would induce changes in the fatigue distribution that could be challenging for the method to handle. For example if severe resonance effects are encountered. On the other hand, a support structure design is usually constructed with a specific turbine model in mind and the results have shown that the method can handle significant changes to global dynamics to a certain extent (as seen for model MI10 in Fig. 4). Furthermore, severe resonance is hardly desirable in any case and such designs would likely be ruled out by other means. Hence, while we recommend that the method be trained for use with only a single turbine model at a time, as long as the impact on global dynamics is not too significant, the method could still be viable for related turbine models within somewhat relaxed error criteria.

Another possible limitation, at least for some applications of the method, is that the results here have been derived using only normal stress. On the one hand, this is standard practice in the industry and therefore also for many research applications. Furthermore, the methodology has for some time been seen to give fairly accurate (often conservative) fatigue estimates for applications in the oil and gas industry (see e.g. Lotsberg (2016)). On the other hand, there are certainly cases where multiaxial stress is important to consider. However, the procedure required to account for this is quite involved. Calculating multiaxial stress requires the use of shell elements rather than beam elements. This makes the modeling and time-domain analysis much more complicated than what has been done in this study. Additionally, the estimation of fatigue damage from multiaxial stress is also more complicated and less standardized, in particular the cycle counting (see e.g. Stephens (2001)). Hence, we consider the effect of multiaxial stress to be outside the scope of this study and would therefore advise caution when using the method for such applications.

## 4.2  Applications to design optimization and preliminary design

One of the most discernible outcomes of the testing framework is the indication that the method works best for designs that have been randomly modified, as seen when comparing Fig. 6 with Fig. 4. As noted previously, and especially evident for design MI10, systematic changes in the structure will to a larger extent cause changes in global dynamics that decrease the performance of the method. Specifically, the method relies on proportional changes in the fatigue damage across all load cases. As we have seen, this property is sensitive to, e.g., changes in eigenfrequency. Random changes to the structure have a much smaller impact on the eigenfrequencies and other global phenomena that are expected to skew the fatigue damage distribution across all load cases. Since random changes more closely resemble the configurations most relevant for design optimization, the method seems very promising for this application. While it can occur that large systematic changes result from an optimization loop, e.g. if the original structure is significantly over- or under-designed with respect to fatigue resistance, most of the computational work in most cases will occur in stages where the overall changes to the structure are small. One can certainly also envision applications of this method to preliminary design, where perhaps a larger extent of the work is in rough scaling of the design. In most cases, the errors reported here are small enough also for these design situations. Even the larger errors reported (in the case of 10% up-scaling) might be acceptable in the early phases of design.

The various design configurations that were used to test the method were chosen in an attempt to cover as many scenarios of interest as possible. However, not all types of scenarios could be accommodated and hence there are some configurations about which we cannot make strong conclusions. The most obvious of these is the fact that we have scaled both diameters and thicknesses by the same factor. Even for the randomized designs, the scale factor was only randomly sampled on an element-wise basis. A situation where either diameters are increased and thicknesses decreased or vice versa, could easily occur in practice during design optimization. On the other hand, based on our result, it seems that the most significant factor in determining the effectiveness of the method is whether or not there are global changes in eigenfrequency. Hence, though we are unable to explicitly confirm this based on our results, we expect that even in configurations like that described above (or other potential untested ones), the method should be viable under the same criteria: As long as there are no global changes that induce non-proportional changes in fatigue damage for only a certain subset of load cases, the method performs well.

### 4.3 Comparison with previous work

Comparing the approach taken in this study with most previous work on load case reduction, certainly the studies cited in the introduction of this paper, one of the main advantages is the simplicity of the method. Because most of the other studies (e.g. Häfele et al. (2018) and Müller and Cheng (2018)) have slightly different aims, i.e. reducing the number of load cases for single design situations, it is not necessarily sensible to compare directly the achieved accuracy for a given amount of load case reduction (though if one were to do so, it would be a reasonably favorable comparison). Something similar could be argued in terms of the methodology, that such a simple approach is only possible in the current setting, but we would still stress the overall simplicity as a major reason why this method would be useful. Especially the avoidance of more advanced statistical and computational procedures (like in Müller and Cheng (2018) and Kim et al. (2018)), will likely make this approach more appealing for industrial applications. There is also little reliance on software, requiring only the ability to sort the fatigue data and then create sampling sets where duplicate load cases have been removed. Furthermore, we note that since the method is completely deterministic (as opposed to many sampling-based approaches), there is little or no uncertainty in the results reported here. In other words, while the specific results (say whether $k$ samples gives an error of exactly $x\%$) are tied to specific background details of the study (the models used, the load case data, etc.), if the method gives a certain accuracy for a certain set of data, it will always give this accuracy for that data.

### 4.4 Possible continuations

The simplicity of the method might also suggest the possibility of improvements, at least in some of the scenarios shown. While some attempts at applying sequence acceleration techniques were made, with little or no positive effects (hence why this was not shown), it is certainly possible that such approaches, or similar ideas, might decrease the error of the estimates or at least decrease the number of samples needed to reach a certain level. We additionally note that further ideas for how to apply the method for specific applications could also be developed. For example, since systematic design modifications of a certain size can impact the accuracy of the method, as seen especially for design MI10 in Fig. 4, it would be possible to apply the method in an adaptive way for, e.g., optimization. One could argue that such adaptive strategies are not necessary, since

it is often possible to avoid such inaccuracies by enforcing eigenfrequency constraints. However, if it is known a priori that certain changes in the eigenfrequencies can decrease the performance of the method due to dynamic amplification for some wind speeds, then one possible adaptive strategy would be to implement a check for this situation which when triggered would have an effect on how the method was utilized. When such large global changes to the structure would be detected, one could for example either increase the number of samples used or perhaps require a new full analysis to update the data used to train the method. In other words, such an adaptive strategy would define a kind of "safe" region of design configurations in which the method could be applied very accurately (somewhat analogous to trust region methods in mathematical optimization, see e.g. Nocedal and Wright (2006)) and would change the way the method was applied whenever the design was no longer in this region. One can also envision other types of applications, where something other than (or at least not exclusively) the design is modified. For example probabilistic design/reliability analysis, where the statistical behavior under the variation of a set of input parameters is investigated. While this would have to be verified in a separate, future study, one can envision the method being employed in a similar fashion as here: Training the method on a base parameter configuration and then reducing the number of load cases needed for fatigue assessment when the parameters are allowed to vary.

One limitation of the results obtained in this study is the fact that only operational loading conditions (power production) were analyzed. Since many other conditions are relevant for design, it would pertinent to ask whether the method could be extended to these cases as well. Based on the results obtained here, it seems clear that the effectiveness of the method in these other scenarios would depend on whether or not the fatigue damage also in these cases changes proportionally when the design is modified. If this property still holds, then most likely the error level when using only the $k$ most severe load cases would still be approximately invariant and the method should work fairly well. If this property does not hold, the accuracy of the method could be significantly reduced. Investigating the performance of the method for other types of load cases would be an interesting continuation of the present study.

## 5    Conclusions and outlook

In this study we have presented a simple approach for reducing the number of load cases required for accurate fatigue assessment of an offshore wind turbine support structure under operational conditions. By making a simple assumption about the relative error incurred by only using the most severe load cases in the total fatigue sum, specifically that this error remains approximately constant as the design is modified, we are able to make accurate predictions for the fatigue damage of a set of seven modified designs. One key part of the method is that the ordering of the severity of each load case is slightly different from location to location. Hence, we have used the union of the reduced sets at each location to form a total sampling set that is used in the method. While slightly increasing the number of samples needed, this has a significant impact on the overall performance in terms of balancing the accuracy at each location in the structure. The overall results of the method are very promising, achieving errors of a few percent or less for sample sizes of 15-60, depending on how the designs have been modified. Only in one case, where the increased dimensions of the design caused significant changes in the eigenfrequency and subsequent dynamic amplification for some wind speeds, were the errors a bit higher. Though still in this case less than 6% for comparable

sample sizes. Considering that even a sample size of 100 means a reduction of the load case set (initially numbering 3647) of about a factor of 36, the method generally allows for very large savings in computational effort for fatigue assessment. The method is particularly effective for designs where modifications have been made randomly from element to element, achieving errors of less than 1% for reasonably small sample sizes. This in particular, though also the overall performance, makes the method useful for applications to design optimization. The fact that the method seems to consistently under- or over-predict the fatigue damage based on whether the design has been consistently scaled up or down even makes it possible in some situations to further correct the estimates in order to ensure that the method is always conservative.

One clear advantage compared to state-of-the-art approaches for load case reduction, aside from the overall accuracy, is the simplicity of the method. Whereas the most common approaches rely on various types of sampling techniques that require some amount of statistical and computational complexity, our approach relies entirely on sorting, the union of small sets (combining and then discarding duplicates) and basic arithmetic. Aside from the overall attractiveness of such simplicity, this makes the method more useful for applications in industry where complex methodologies can lead to unacceptable bottlenecks in the work flow. The simplicity of the method presented in this study (on both a conceptual and implementation level) could also be attractive for other scientists, who may not be as comfortable with advanced sampling methods.

While the method as is can readily be applied in many settings, some future developments can be envisioned. For example, one could study possibilities for improving the convergence of the estimates or investigate specific ways of applying the method to design optimization that adapts to regimes where the estimates are expected to lose accuracy. A future study might also look into whether, or to what extent, the method could be extended for use within a probabilistic design or reliability framework. In practice, this would mean seeing whether the fundamental assumption of the method, the invariance of the relative fatigue estimation error when sampling only the most severe load cases, also holds when parameters other than those related to the structural dimensions are altered. Finally, the performance of the method for other support structure types (jackets, floating support structures, etc.), other turbine models and other loading scenarios (other than power production) are all open questions for future work.

*Code and data availability.* The data used for plotting the figures, and corresponding python scripts to make the plots, are available as supplementary material. The raw fatigue data is available upon request. The underlying raw simulation data is too large to distribute.

*Competing interests.* The authors declare that they have no competing interests.

*Acknowledgements.* This work has been partly supported by NOWITECH FME (Research Council of Norway, contract no. 193823) and by the Danish Council for Strategic Research through the project "Advancing BeYond Shallow waterS (ABYSS) - Optimal design of offshore wind turbine support structures".

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
