# Peer review of "Reducing the number of load cases for fatigue damage assessment of offshore wind turbine support structures by a simple severity-based sampling method"

_Wind Energy Science, 2018_

## Referee Comment (RC1) · J. Häfele (Referee) · 13 Jun 2018

This manuscript addresses a common problem in structural design of structures for offshore wind turbines, where the computational costs for FLS structural code checks are high. In general: Good work! The proposed approach is straightforward and the paper is well-written. I also believe that this work is relevant to practical applications. From my point of view, the following points can be strengthened or discussed, respectively:

- Page 2, line 2: "Furthermore, a fundamental assumption for this method is that

the relative fatigue response to each load case remains approximately constant for an extended family of related support structure designs" - This is indeed a fundamental assumption and it is shown that it is valid under the given boundaries for the given (NREL 5MW) turbine. However, it is important to highlight that this may be invalid for a different turbine (due to severe resonance effects, for instance).

- Subsection 2.4: Needs (minor) improvement concerning description of load assumptions, i.e., how does your wave spectrum look like or how do you model the current?

- Subsection 2.4: Can you elaborate a bit more on your "elements" or your structural model, respectively? I am actually not familiar with Fedem and I guess I am not the only one, so can you provide some more details?

From my point of view, the manuscript can be recommended for publication, when these points have been addressed.
Some minor remarks:

- It may increase the quality of the paper, when you use the same font style in all figures.

- Page 8, line 4: "has been quantified"?.

- Page 14, line 27: "state-of-the-art approaches".

- In your references list, try being consistent: Either "Jason Jonkman" or "Jason M. Jonkman".

- References from DNV GL: Particularly the first one is antiquated. Take these: https://rules.dnvgl.com/docs/pdf/DNVGL/RP/2016-04/DNVGL-RP-C203.pdf (RP-

C203), http://rules.dnvgl.com/docs/pdf/DNV/codes/docs/2016-04/Os-J101.pdf
(OS-J101).

---

## Referee Comment (RC2) · Anonymous Referee #2 · 10 Jul 2018

General comments The authors have proposed a method to decrease the number of load cases to be evaluated during fatigue analyses in wind turbine support structures. The method is interesting and could be very useful especially for optimization applications. However, some comments and suggestions are given below with the aim of clarifying the advantages and possible limitations of the method as well as to improve the quality of the document itself.

Specific comments P1. L11-L12. I would suggest rewriting the sentence "The method as is can be used without further modification" because it sounds like the method can-

not be improved and there is always a possibility of improvement. P2. L6-L7. What would be the effect of considering other design situations besides the power production, such as parked conditions? I would suggest adding a short clarification about this. P3. L22. Considering normal stresses means that the damage is estimated assuming under uniaxial stress states. How real is this assumption for these type of structures which are normally subjected to multiaxial stress states? What would be the effect of considering multiaxial stress states in the proposed model?

P3. L26-L27. Do the authors mean: the maximum value of the total damage among the eight points after evaluating all possible load cases? If so, make a clarification.

Regarding Fig. 1-b Does the Normalized fatigue damage correspond to $D_k/D_{tot}$ ? If so, add clarification in the figure. How was the proportion of total load cases calculated? How Fig. 1-b would look for the different evaluated points along the tower?

P4. L11. What does it mean "small" and "intermediate" values of k? How is that scale defined? P6. L18. How many random seeds were used for each load case in this study? What would be the effect of the number of seeds on the final number of load cases to be evaluated?

P6. L28-L31. Could the authors elaborate more about how was the scaling process of the element sizes carried out? Were the element sizes scaled only once or several times until the optimal solution was found?

P7. L3-L6. The statement "From the distribution shown….." is not clear from Fig. 2-a. In this figure, no wind speeds are shown but load cases, which are not clear either. In addition, how can be proved that the load cases with highest normalized fatigue damage are those having the highest probability of occurrence? Is there any reference or way to show this? What does it mean "Normalized fatigue damage"? If you want to show the level of severity, why are you plotting the normalized fatigue damage instead of severity level? I would suggest explaining better this figure both in the figure itself and in the text.

P8. L10. Regarding the statement "However, this turns out to not be the case.", is this statement for this specific case or in general? If it were for this specific case, what would be the consequences on the proposed model in those cases when the sampling sets are much larger than the number of load cases at each location? If it were in general, how can you prove this statement?

P8. L14-L16. It would be good to show Fig. 2-b for the three evaluated points. That would provide more veracity to the statement given in this paragraph.

P9. L25-L29. Regarding the statement, "We observe that the method seems to consistently over-predict the fatigue damage…" What is the consequence of this? Could there be cases in which the results obtained by the method can lead to under-estimated designs (which are not desirable in any structural design)?

Regarding Figures 3 and 4. If the error can have both negative and positive values, it means that the estimated damage value could be greater than the real damage value. How could be that possible? So, how would you choose the optimal sample set size which makes a balance between the number of loads to evaluate and the final accuracy? Would it be possible to find this value by implementing a simple optimization process? How is the behavior after 180 load cases? It would be good to show more results taking into account that the real number of load cases is larger than 3000. In this way, you could show with more confidence the accuracy of the model.

P9. L6-L7. Regarding the statement, "This in turn makes…" What would be a possible solution for this?

P10. L7-L12. Not sure how pertinent is this discussion for the purpose of this paper.

According to this section 3.3., the level of accuracy of the proposed model could decrease considerably when many points in the structure are analyzed since the sample set size could much higher than the number of load cases at each point (i.e. n»k). How could this limitation be controlled? This is especially important when the entire structure is analyzed under fatigue. Regarding Fig. 5, Add the location of the point along the tower at each plot of the figure. How is the error shown in Figure 5 for a greater number of load cases, e.g. 200, 500, 1000?

Regarding section 4.1, I would suggest analyzing the viability and the limitations of the proposed model in a general point of view instead of focusing only in the evaluated optimization methodologies (i.e. MD5, MD10, etc.). The readers might have other optimization methodologies and it would be useful for them to know when they can implement this method. P12. L10 to P13. L3. Elaborate more on these statements, they are not clear as they are now. P13. L6-L9. This is an honest and significant statement. P13. L19-L23. I do see important to consider in future works the uncertainty related to the chosen number of load cases k and, even more, the one related to the final sample set size n. It would be good to add a diagram summarizing the proposed model. I did not find any comparison or references to previous works during the discussion.

Technical comments P1. L13. Change "a few" for "some" P1. L15-L18. The two first sentences (i.e. "A central practical. . .." and "In order to assess..") could be rewritten in a shorter and clearer way. P1. L21. Commission P3. L13-L17. I would suggest deleting this paragraph. This information is not necessary. P3. L19. It is not clear what the authors mean in the first sentence. Rewrite it. P5. L2. ". . ..of some new designs of the same structure, with. . .."

P6. L5-L7. Write the last sentence of this paragraph also in equations. That would make the idea clearer.

Regarding Fig. 2. Add a legend defining both the green points and the blue points What is the x-axis scale?

P7. L3. Change "in the left panel of Fig. 2" for "Fig. 2-a"

P8. L3. Make clearer which type of design is refereeing in "For each design,. . .".

P8. L4. "Specifically, the performance of the method has been quantified. . ."

Regarding Eq. 7, define the variable in the text.

P8. L11. Change "...in the right panel of Fig. 2." for "Fig. 2-b".

P8. L11-L12. "It is reasonably linear, varying between n=7 for k=7 and n=181 for k=150."

P9. L18. "The relative errors $\delta$(?) for various sample sizes n (?) is shown...."

P9. L18. Are the "designs" in this line related to the "new designs" mentioned in P4. L2.?

Regarding Figures 3 and 4. Add the name of the models (i.e. MD5, MI5, etc.) at each plot of these figures in order that each plot can be understood itself without the need for the reader to read the caption of the entire figure. Is the "Relative error" at the y-axis referred to $\delta$ from Eq. 7? If so, add $\delta$ in the y-axis, as well as the units.

P9. L3. Define "3P frequency"

P9. L9. "The relative errors $\delta$(?)"

P10. L9. Double "observe" P10. L18. Use different notations for the value Ït_kshown in Eq. 3 and the actual one. Regarding Eq. 8, define in the text all variables of this equation. P12. L10. Change "given that" for "because". P13. L3. I would say "With regards to applications to design optimization, this method seems to be very promising". P13. L10. I would eliminate this title and add section 4.3 to section 4.2. P13. L28-L30. Elaborate more on this idea and change "(e.g.)" for ", e.g.," or ", for example,". The text is full of informal language, like the ones shown below. I would suggest using a more formal language (e.g. In other words, shown, etc.). Substantial machinery in place (P3. L19.) Effectively speaking (P5. L5.) In plain words (P5. L21.) That is to say (P7. L7.) this turns out (P8. L10.) displayed (P10. L18.) Put in another way (P13. L21.)

---

## Editor Comment (EC1) · L. P. Mikkelsen (Editor) · 23 Aug 2018

Thanks for your response to the referee comments. Looking forward to the updated version of the manuscript. Please remember to mark by color the changes you have made in the manuscript and indicated in your author response to the referee how you have addressed each of the point raised by the reviewers. This will make it much easier for the two reviewers to judge whether all their points have been addressed. It is preferable to use a color marking instead of just use the track-changes feature in e.g. MS-word, as it in the case of many corrections will make the manuscript difficult to

read.

---

## Author Comment (AC1) · 23 Aug 2018

**Author response to interactive comments on "Reducing the number of load cases for fatigue damage assessment of offshore wind turbine support structures by a simple severity-based sampling method"**

Lars Einar S. Stieng          Michael Muskulus

First and foremost, we wish to offer a general thanks to both referees for carefully reading and offering constructive criticism of our manuscript. Their feedback will help to improve the quality of the work and of the writing.

In the following, statements by the referees have been italicized.

**Response to Referee #1, Jan Hëfele**

*This manuscript addresses a common problem in structural design of structures for offshore wind turbines, where the computational costs for FLS structural code checks are high. In general: Good work! The proposed approach is straightforward and the paper is well-written. I also believe that this work is relevant to practical applications.*

Our sincere thanks to Jan Hëfele for taking the time to read our manuscript and for offering some useful advice for improvements. The feedback is highly appreciated!

*Page 2, line 2: "Furthermore, a fundamental assumption for this method is that the relative fatigue response to each load case remains approximately constant for an extended family of related support structure designs" - This is indeed a fundamental assumption and it is shown that it is valid under the given boundaries for the given (NREL 5MW) turbine. However, it is important to highlight that this may be invalid for a different turbine (due to severe resonance effects, for instance).*

This is indeed true. The response of the structure is highly dependent on the turbine, both in terms of the loads and the overall dynamics induced by the turbine (1P and 3P frequencies for instance). One could argue that since any support structure must be designed with this in mind, any design that comes close to exhibiting severe resonance effects would likely be ruled out by other means (in optimization, constraints on allowable

values for the first eigenfrequency are common). Some resonance effects were already seen in the results (when the design was scaled up by 10%) and the effect on the method was noticeable if not too severe. All in all this means that the tolerance for such effects within the method is enough that it most works (with some reduced efficiency), at least up to a point where the design being studied would still be viable under a more general set of criteria. All that being said, we agree that this point could have been stated more explicitly in the text and we will update accordingly in the revised version.

*Subsection 2.4: Needs (minor) improvement concerning description of load assumptions, i.e., how does your wave spectrum look like or how do you model the current?*

*Subsection 2.4: Can you elaborate a bit more on your "elements" or your structural model, respectively? I am actually not familiar with Fedem and I guess I am not the only one, so can you provide some more details?*

These details were originally omitted for brevity, but we agree that some readers may find them of interest. Especially for the sake of reproducibility. A more detailed description of these details will be added in the revised version.

*It may increase the quality of the paper, when you use the same font style in all figures.*

We have tried to be consistent with details like these where possible. However, it may be possible to improve consistency a bit more with a second look. If possible, this will be addressed in the revised version.

*Page 8, line 4: "has been quantified"?.*

*Page 14, line 27: "state-of-the-art approaches".*

*In your references list, try being consistent: Either "Jason Jonkman" or "Jason M. Jonkman".*

*References from DNV GL: Particularly the first one is antiquated. Take these: https://rules.dnvgl.com/docs 04/DNVGL-RP-C203.pdf (RPC203), http://rules.dnvgl.com/docs/pdf/DNV/codes/docs/2016-04/Os-J101.pdf (OS-J101).*

Thank you for pointing these out. They will be fixed/updated in the revised version.

**Response to Referee #2**

*General comments The authors have proposed a method to decrease the number of load cases to be evaluated during fatigue analyses in wind turbine support structures. The method is interesting and could be very useful especially for optimization applications. However, some comments and suggestions are given below with the aim of clarifying the advantages and possible limitations of the method as well as to improve the quality of the document itself.*

Our sincere thanks to Referee #2 for the careful reading of and comprehensive response to our manuscript. The comments will serve well to improve the quality of the paper. The feedback is highly appreciated!

*P1. L11-L12. I would suggest rewriting the sentence "The method as is can be used without further modification" because it sounds like the method can not be improved and there is always a possibility of improvement.*

The intention with this sentence was to convey the sense that no further improvement was strictly necessary before the method could be applied, but of course there is always room for improvements in one way or another. It certainly was not our intention to imply otherwise. We appreciate that the way it was written is not completely clear on this point and we will rephrase the sentence in the revised version to make our intended meaning more explicit.

*P2. L6-L7. What would be the effect of considering other design situations besides the power production, such as parked conditions? I would suggest adding a short clarification about this.*

Since we have not attempted this in our study, we cannot say for sure. However, it would most likely depend on how sensitive the fatigue damage in these design situations is to changes in the design. Or, specifically, whether or not the fatigue damage in these cases changes in ways that are proportional to the changes for power production. If the changes are proportional, then the method would still work as reported here. If not, then the results would be weakened, though it is hard to say by how much. We agree that, although the method has been designed specifically to reduce the very large computational effort associated with analyzing the power production DLC, a clarification of this point would be both instructive generally speaking and could also point the way for further studies. A short statement about this will be added in the revised version.

*P3. L22. Considering normal stresses means that the damage is estimated assuming under uniaxial stress states. How real is this assumption for these type of structures which are normally subjected to multiaxial stress states? What would be the effect of considering multiaxial stress states in the proposed model?*

It is true that these structures are in reality subject to multiaxial stress. However, it has long been standard practice in the industry, and hence also in a lot of research aiming for industrial applications, to consider normal stress only. The main reason for this is that it allows the use of beam models and therefore a much less involved structural analysis when calculating stress and fatigue damage. To properly calculate multiaxial stress means having to use shell elements instead of beam elements in the finite element analysis, which requires more effort both on the modeling side and on the computational side. Additionally, the calculation of fatigue damage from multiaxial stress is much more involved, especially when it comes to the choice of methodology for identification and counting of stress cycles (see e.g. Stephens 2001, Metal fatigue in Engineering, for more on the multiaxial approach). Based on decades of experience from the oil and gas industry (see e.g. Lotsberg 2016, Fatigue design of marine structures) it has been seen that the normal stress approach, while not as accurate, gives reasonable (and often

conservative) estimates for the fatigue damage in marine structures. While it would certainly be valuable to understand more about the behavior of the structure under multiaxial stress and how it would affect methods like the one proposed in this paper, especially since the more involved calculations in this case would be an even higher incentive to simplify the load case analysis, this falls outside of the scope of the present work. Without any data to base it on, any notions about how multiaxial stress might affect the effectiveness of the proposed methodology (e.g. more local behavior or larger dependence on directionality) would be highly speculative. However, all that being said, it would be instructive for the reader to understand the possibility of limitations of the method for applications to analysis based on multiaxial stress calculations. We will therefore make it more clear in the revised version that the presented results are in principle only valid for approaches based on normal stress calculations and that the effectiveness of the method could be reduced if used with multiaxial stress analysis.

*P3. L26-L27. Do the authors mean: the maximum value of the total damage among the eight points after evaluating all possible load cases? If so, make a clarification.*

No. This is done per load case. For each load case, the stress is calculated at eight points around the circumference of a given location in the structure (e.g. mudline). The fatigue damage resulting from each stress value is compared and the largest one is selected to represent the fatigue value of this location in the structure, for this specific load case. We can see how the original phrasing of this was a bit unclear and will re-write accordingly in the revised version to make sure the intended meaning comes across more clearly.

*Regarding Fig. 1-b Does the Normalized fatigue damage correspond to $D_k/D_{tot}$? If so, add clarification in the figure. How was the proportion of total load cases calculated? How Fig. 1-b would look for the different evaluated points along the tower?*

Yes, the partial sums were normalized to the total sum. We agree that making this more explicit also in the figure would be instructive and will do so in the revised version.

The proportion of total load cases was calculated as the number $k$, the number of load cases used to calculate the k-th partial sum, divided by the total number of load cases considered (in this case 3647). This could also have been more clearly stated in the figure/caption. The revised version will amend this.

The equivalent curve for the two other points along the tower would be essentially identical and including them in the plot would give little or no additional information. However, this could have been noted in the text so it is more clear that we are not cherry picking the data. The revised version will reflect this.

*P4. L11. What does it mean "small" and "intermediate" values of k? How is that scale defined?*

This wording is indeed imprecise. What was meant was the values of k for which the curve in the figure is seen to approach very closely to the asymptote. The language here was perhaps too reliant on a qualitative judgment and should have been quantified more clearly, with reference to the figure. This will be added in the revised version.

*P6. L18. How many random seeds were used for each load case in this study? What would be the effect of the number of seeds on the final number of load cases to be evaluated?*

For the sake of testing this method, only one seed per 10 minute time series (with given wind speed, sea state and direction for the incoming waves) was used. To conform with standards, a minimum of six seeds (or equivalently, one 60 minute time series) per load case should have been used. However, the use of additional seeds would only serve to stabilize the fatigue values per load case against random fluctuations. If there would be noticeable effect on the results, it would hence be a beneficial one. On the other hand, this information could very well be of interest to the reader and should be stated in the text. A short clarification of this will be added in the revised version.

*P6. L28-L31. Could the authors elaborate more about how was the scaling process of the element sizes carried out? Were the element sizes scaled only once or several times until the optimal solution was found?*

The elements were scaled just once to obtain each of the models used for testing the method. No further scaling was done. In fact, though each model was meant to "simulate" an optimization process, no actual design optimization was done at any point of this study. Clearly our explanation of the testing setup was not clear enough about what was being done. The description here will be updated/expanded in the revised version to more clearly convey the testing methodology.

*P7. L3-L6. The statement "From the distribution shown. . ..." is not clear from Fig. 2-a. In this figure, no wind speeds are shown but load cases, which are not clear either. In addition, how can be proved that the load cases with highest normalized fatigue damage are those having the highest probability of occurrence? Is there any reference or way to show this? What does it mean "Normalized fatigue damage"? If you want to show the level of severity, why are you plotting the normalized fatigue damage instead of severity level? I would suggest explaining better this figure both in the figure itself and in the text.*

To your first point, the load cases were grouped according to wind speed, with the smallest wind speed to the left and the largest wind speed to the right (as stated in the figure caption). However, this could have been more clear from simply looking at the figure itself. At least, each wind speed bin could have been marked on the x-axis and the extent of each bin could have been delineated more clearly. We will update the figure in the revised version to make this information more clearly deducible from the figure itself.

The rest of your points here follow from an unfortunate error on our part which resulted from not being consistent with our own terminology. What is being plotted here is really the severity, normalized to the largest value. Hopefully this explains the rest as well. For example, the fact that the load cases with the highest severity for a given windspeed are the ones with the highest probability of occurrence is simply a direct observation that we have made when comparing the index of the peaks in each bin with the scatter diagrams used. Displaying this information explicitly in the plot would be very difficult and it was simply meant as a small observation. However, we should have been consistent with the use of the term severity also in the figure labels and captions. This will be fixed in the revised version.

*P8. L10. Regarding the statement "However, this turns out to not be the case.", is this statement for this specific case or in general? If it were for this specific case, what would be the consequences on the proposed model in those cases when the sampling sets are much larger than the number of load cases at each location? If it were in general, how can you prove this statement?*

In a certain sense, we can of course only verify this statement as far as the loading conditions and support structure models used in the study are concerned. We do not believe there is anything particularly special about the setup in a way that would make it simplify this behavior compared to other setups, but cannot prove this in practice. If this behavior breaks down, such that the load cases selected from each location were generally not the same, then the efficiency of the method would reduce by up to a factor 3 (in this case, more generally a factor equaling the number of locations). In order to not make it seem like we were making very general conclusions from these results, we should probably have written something like "our results indicate that this is not the case." A few words about the consequences if this result breaks down for other setups could also be added in the discussion section. The revised version of the paper will include these changes.

*P8. L14-L16. It would be good to show Fig. 2-b for the three evaluated points. That would provide more veracity to the statement given in this paragraph.*

Figure 2-b does in fact include all three points. The horizontal axis is the number of load cases taken from each point and the vertical axis is the corresponding total size of the sampling set. That is selecting the, e.g., 20 most severe load cases from each point results in a total sample set size of around 25, and so on. Hopefully this does indeed provide the desired veracity to the statement in the text. However, perhaps this could have been made in even more clear in the text and/or the figure caption. The revised version will be updated accordingly.

*P9. L25-L29. Regarding the statement, "We observe that the method seems to consistently over-predict the fatigue damage. . ." What is the consequence of this? Could there be cases in which the results obtained by the method can lead to under-estimated designs (which are not desirable in any structural design)?*

It is indeed possible that the method can in some cases under-predict the fatigue damage of a given design. However, this under-prediction would then only be at the reported error level. From a more practical point of view, there is of course a large difference between an error that leads to under-prediction (unsafe) and an error that leads to over-prediction (safe). Any method like this would have some amount of error, but the difference is that in our case we see that the tendency to over-predict or under-predict the fatigue damage is highly correlated with how the design has been changed compared to the reference design. It would hence be possible for a designer to apply a safety margin in the case where there would be a strong indication that there could be a slight under-prediction of the fatigue damage. We tried to explicate this fact in the discussion section (Page 12, lines 3-7), but could perhaps have been even more clear in our discussion about these ideas. The revised version will be updated accordingly.

*Regarding Figures 3 and 4. If the error can have both negative and positive values, it means that the estimated damage value could be greater than the real damage value. How could be that possible? So, how would you choose the optimal sample set size which makes a balance between the number of loads to evaluate and the final accuracy? Would it be possible to find this value by implementing a simple optimization process? How is the behavior after 180 load cases? It would be good to show more results taking into account that the real number of load cases is larger than 3000. In this way, you could show with more confidence the accuracy of the model.*

The estimated fatigue value is simply a scaling of the fatigue value of the initial design. If the scale factor is too large, then the estimated damage value would be larger than the real damage.

In a certain sense, we have left the choice of "optimal" sample set size to the reader. We could have set a target accuracy and simply increased the sample set size until this was reached. However, such a target value would be entirely application specific. Some might be happy with an error of 10%, others might want an error less than 1%. Furthermore, the exact sample set size is likely highly dependent on the support structure design, turbine model and loading scenario. While we expect based on our results that sample set sizes of 30-50 would give errors of only a few percent or less (except for designs that have been altered to a large extent), giving an explicit number would limit the results to only the specific case(s) studied.

We terminated our figures at sample set sizes where the error started to flatten out. We also had in mind that for the method to be considered worthwhile, at least when compared to other methods proposed in the cited previous studies, the number of load cases used needed to be small when compared to the total number of load cases (3647). However, we do see the utility of including larger sizes, at least up to reductions of a factor of 10 (approximately 360 load cases), or even 5 (approximately 720 load cases). This would show more clearly the convergence of the results (indicating that including more load cases gives little additional reduction in error). Perhaps such extended curves would also make it easier for the reader to choose their own "optimal" sample set size. We will include this additional information in the figures in the revised version.

*P9. L6-L7. Regarding the statement, "This in turn makes. . ." What would be a possible solution for this?*

The most straightforward solution to this issue is to make sure that the changes in the eigenfrequency of the structure are within a certain tolerance. Specifically, that the structure does not approach regions of significant dynamic amplification for certain windspeeds (as indicated by the Campbell diagram of the turbine model). If such a situation is reached, one would either have to "restart" the method by doing a new full fatigue analysis or one would have to apply a safety factor to ensure that the increased uncertainty would not lead to unsafe designs. This was discussed in the discussion section (Page 13, lines 29-33 and Page 14, lines 1-2), but could have been more explicit in its connection with specific results. This discussion will be expanded in the revised version.

*P10. L7-L12. Not sure how pertinent is this discussion for the purpose of this paper.*

This discussion is meant to illustrate how knowledge about how the method behaves under significant systematic changes can be used to improve the accuracy, or at least the safety, of the predicted fatigue damage. Together with the later discussion, as referenced previously, this allows the reader to understand how (e.g.) safety factors can be applied to ensure that fatigue predictions never lead to under-designed structures.

*According to this section 3.3., the level of accuracy of the proposed model could decrease considerably when many points in the structure are analyzed since the sample set size could much higher than the number of load cases at each point (i.e. n¿¿k). How could this limitation be controlled? This is especially important when the entire structure is analyzed under fatigue.*

Here it seems we have not been clear enough in our explanations. The absolute level of accuracy does increase, it is merely that the relative accuracy does not. There are two reasons for this. One is that as $\epsilon_k$ and $\epsilon_k^{\text{new}}$ become very small for larger values of $k$, the relative accuracy becomes less meaningful (a relative difference of 100% at numerical values of $10^{-8}$ is hardly a significant error in practice). The second, and perhaps more convincing, reason is that while both parts of the fraction in Eq. 8 do tend toward zero, the denominator will tend to zero faster. This does not mean that accuracy is decreasing for higher values of $k$, it merely means that there is little gained for the method in general once $k$ increases past a certain point. The convergence towards the exact answer is very slow, as each additional load case gives very little new information. All in all, this could have been more clearly written and explained in the text. The revised version will be updated to reflect this.

As for the effect of including more points, we think that having selected three points at large separation in the structure, and having shown from Figure 2-b that the number of additional load cases needed in the sample set is fairly small, that the results obtained would indicate that there should be little additional computational effort or loss of accuracy if the entire structure is analyzed.

*Regarding Fig. 5, Add the location of the point along the tower at each plot of the figure. How is the error shown in Figure 5 for a greater number of load cases, e.g. 200, 500, 1000?*

The location along the tower of each subplot, though indicated in the main caption, could certainly be included in each sub-caption.

As discussed above, the relative error shown in Figure 5 will in fact increase for a larger number of load cases. While some additional information could have been included here, the values of $k$ selected were meant to show the behavior at the values determined most relevant from the previous results. However, to illustrate this point a bit more clearly, we could certainly add a few more rows (for example for the suggested values of $k$) to the figure in the revised version.

*Regarding section 4.1, I would suggest analyzing the viability and the limitations of the proposed model in a general point of view instead of focusing only in the evaluated optimization methodologies (i.e. MD5, MD10, etc.). The readers might have other optimiza-*

*tion methodologies and it would be useful for them to know when they can implement this method.*

Referring back to our reply regarding the explanation of the testing setup, we would again stress that the different structural models analyzed (MD5, MD10, etc.) do not constitute different optimization methodologies, but rather different, fixed states of the structure that would likely be encountered during many types of optimization contained within the relevant scope of the application (mass/weight optimization of the support structure). Of course we cannot claim to have covered everything and we could perhaps have discussed some possible limitations related to this, but overall we think that focusing precisely on the behavior of these test scenarios represent the best way to draw a more general set of conclusions about how the method might behave in different situations. Some further discussion to underline how these models represent (or do not) relevant scenarios that could motivate the reader to use the method for their own purposes will be added in the revised version.

*P12. L10 to P13. L3. Elaborate more on these statements, they are not clear as they are now.*

This was meant to also refer back to some of the discussion of the results in the previous section, but it was clearly written either too briefly or not clearly enough. This will be updated in the revised version.

*P13. L19-L23. I do see important to consider in future works the uncertainty related to the chosen number of load cases k and, even more, the one related to the final sample set size n.*

This is true, though it is hard to do so without performing a rather large and comprehensive comparison of different support structures, different turbine models and different environmental data.

*It would be good to add a diagram summarizing the proposed model.*

A flow chart or similar that summarizes the steps involved in the method could certainly be added in the revised version.

*I did not find any comparison or references to previous works during the discussion.*

We did not make any specific references here, since we already discussed some details of previous work in the introduction and hoped that this would still be on the reader's mind. However, we agree that this makes this subsection a bit hard to read in isolation and that making some more specific references would help make the points more clear. This will be added in the revised version.

*P1. L13. Change "a few" for "some"*

Noted.

*P1. L15-L18. The two first sentences (i.e. "A central practical. . .." and "In order to assess..") could be rewritten in a shorter and clearer way.*

Agreed.

*P1. L21. Commission*

Noted.

*P3. L13-L17. I would suggest deleting this paragraph. This information is not necessary.*

Such a paragraph is standard practice in many cases, but it is not essential. We take this feedback under advisement.

*P3. L19. It is not clear what the authors mean in the first sentence. Rewrite it. P5. L2. ". . ..of some new designs of the same structure, with. . .."*

Noted.

*P6. L5-L7. Write the last sentence of this paragraph also in equations. That would make the idea clearer.*

Noted.

*Regarding Fig. 2. Add a legend defining both the green points and the blue points What is the x-axis scale?*

Noted.

The x-axis has no conventional scale, but is rather a collection of all load case indices (initially not shown because there are 3647 indices). As noted previously, we will make some changes to this figure to make this information more clear.

*P7. L3. Change "in the left panel of Fig. 2" for "Fig. 2-a"*

*P8. L3. Make clearer which type of design is refereeing in "For each design,. . .".*

*P8. L4. "Specifically, the performance of the method has been quantified. . ."*

*Regarding Eq. 7, define the variable in the text.*

*P8. L11. Change ". . .in the right panel of Fig. 2." for "Fig. 2-b".*

*P8. L11-L12. "It is reasonably linear, varying between $n=7$ for $k=7$ and $n=181$ for $k=150$."*

All noted.

*P9. L18. "The relative errors $\delta$ (?) for various sample sizes $n$ (?) is shown. . .."*

It is indeed $\delta$ and $n$ being referred to. This will be added.

*P9. L18. Are the "designs" in this line related to the "new designs" mentioned in P4. L2.?*

Yes. Again, we will make some changes to the description of the testing setup to make

this more explicit.

*Regarding Figures 3 and 4. Add the name of the models (i.e. MD5, MI5, etc.) at each plot of these figures in order that each plot can be understood itself without the need for the reader to read the caption of the entire figure. Is the "Relative error" at the y-axis referred to $\delta$ from Eq. 7? If so, add $\delta$ in the y-axis, as well as the units.*

Noted.

Yes, it is $\delta$ and this will be added. However, since this is a dimensionless number, there are no units.

*P9. L3. Define "3P frequency"*

*P9. L9. "The relative errors $\delta(?)$"*

Noted.

*P10. L9. Double "observe" P10. L18. Use different notations for the value shown in Eq. 3 and the actual one. Regarding Eq. 8, define in the text all variables of this equation. P12. L10. Change "given that" for "because". P13. L3. I would say "With regards to applications to design optimization, this method seems to be very promising"*

All noted.

*P13. L10. I would eliminate this title and add section 4.3 to section 4.2*

We take this under advisement.

*P13. L28-L30. Elaborate more on this idea and change "(e.g.)" for ", e.g.," or ", for example,"*

This idea is in fact continued in the next few sentences of the paragraph, but this could have been written in a more clear way.

Noted.

*The text is full of informal language, like the ones shown below. I would suggest using a more formal language (e.g. In other words, shown, etc.). Substantial machinery in place (P3. L19.) Effectively speaking (P5. L5.) In plain words (P5. L21.) That is to say (P7. L7.) this turns out (P8. L10.) displayed (P10. L18.) Put in another way (P13. L21.)*

Generally speaking, this is a matter of taste. However, there are some examples here that probably push into informal language in a way that may make the text less understandable for some. We will make the necessary adjustments to avoid this. Otherwise, this comment is taken under advisement.

---

## Author Response (AR1)

**Author response to interactive comments on "Reducing the number of load cases for fatigue damage assessment of offshore wind turbine support structures by a simple severity-based sampling method"**

Lars Einar S. Stieng        Michael Muskulus

First and foremost, we wish to offer a general thanks to both referees for carefully reading and offering constructive criticism of our manuscript. Their feedback will help to improve the quality of the work and of the writing.

**Please note that a version of the revised manuscript with marked up changes has been appended at the end of this document.**

In the following, statements by the referees have been italicized and specific references by the authors to updated material in the revised manuscript have been written in boldface.

**Response to Referee #1, Jan Häfele**

*This manuscript addresses a common problem in structural design of structures for offshore wind turbines, where the computational costs for FLS structural code checks are high. In general: Good work! The proposed approach is straightforward and the paper is well-written. I also believe that this work is relevant to practical applications.*

Our sincere thanks to Jan Häfele for taking the time to read our manuscript and for offering some useful advice for improvements. The feedback is highly appreciated!

*Page 2, line 2: "Furthermore, a fundamental assumption for this method is that the relative fatigue response to each load case remains approximately constant for an extended family of related support structure designs" - This is indeed a fundamental assumption and it is shown that it is valid under the given boundaries for the given (NREL 5MW) turbine. However, it is important to highlight that this may be invalid for a different turbine (due to severe resonance effects, for instance).*

This is indeed true. The response of the structure is highly dependent on the turbine, both in terms of the loads and the overall dynamics induced by the turbine (1P and

3P frequencies for instance). One could argue that since any support structure must be designed with this in mind, any design that comes close to exhibiting severe resonance effects would likely be ruled out by other means (in optimization, constraints on allowable values for the first eigenfrequency are common). Some resonance effects were already seen in the results (when the design was scaled up by 10%) and the effect on the method was noticeable if not too severe. All in all this means that the tolerance for such effects within the method is enough that it mostly works (with some reduced efficiency), at least up to a point where the design being studied would still be viable under a more general set of criteria. All that being said, we agree that this point could have been stated more explicitly in the text.

**We have added a paragraph covering this issue in section 4.1 (page 17, line 1 to 10, line 4 in the revised manuscript; page 17, line 22 to 31 in the marked up version).**

*Subsection 2.4: Needs (minor) improvement concerning description of load assumptions, i.e., how does your wave spectrum look like or how do you model the current?*

*Subsection 2.4: Can you elaborate a bit more on your "elements" or your structural model, respectively? I am actually not familiar with Fedem and I guess I am not the only one, so can you provide some more details?*

These details were originally omitted for brevity, but we agree that some readers may find them of interest. Especially for the sake of reproducibility.

**We have added some additional information covering these issues to section 2.4 (page 6, line 16 to page 7, line 6 in the revised manuscript; page 6, line 25 to page 7, line 3 in the marked up version).**

*It may increase the quality of the paper, when you use the same font style in all figures.*

We have tried to be consistent with details like these where possible. However, it may be possible to improve consistency a bit more with a second look.

**The font used within Fig. 1(a) (top of page 4 in the revised manuscript; top of page 5 in the marked up version) was made to conform with the font used in the other figures. Otherwise the font type (and for the most part the font sizes) should be the same for each figure.**

*Page 8, line 4: "has been quantified"?.*

**Done (page 9, line 11 in the revised manuscript; page 9, line 5 in the marked up version).**

*Page 14, line 27: "state-of-the-art approaches".*

**Done (page 20, line 8 in the revised manuscript; page 21, line 1 in the marked up version).**

*In your references list, try being consistent: Either "Jason Jonkman" or "Jason M.*

*Jonkman".*

**Done (page 21, line 16 in the revised manuscript; page 22, line 18 to 19 and line 24 to 25 in the marked up version).**

*References from DNV GL: Particularly the first one is antiquated. Take these: https://rules.dnvgl.com/docs/pdf/DNVGL/RP/2016-04/DNVGL-RP-C203.pdf (RPC203), http://rules.dnvgl.com/docs/pdf/DNV/codes/docs/2016-04/Os-J101.pdf (OS-J101).*

**Done (page 21, line 4 to 5 in the revised manuscript; page 22, line 4 to 7 in the marked up version).**

Thank you for pointing these out.

**Response to Referee #2**

*General comments The authors have proposed a method to decrease the number of load cases to be evaluated during fatigue analyses in wind turbine support structures. The method is interesting and could be very useful especially for optimization applications. However, some comments and suggestions are given below with the aim of clarifying the advantages and possible limitations of the method as well as to improve the quality of the document itself.*

Our sincere thanks to Referee #2 for the careful reading of and comprehensive response to our manuscript. The comments will serve well to improve the quality of the paper. The feedback is highly appreciated!

*P1. L11-L12. I would suggest rewriting the sentence "The method as is can be used without further modification" because it sounds like the method can not be improved and there is always a possibility of improvement.*

The intention with this sentence was to convey the sense that no further improvement was strictly necessary before the method could be applied, but of course there is always room for improvements in one way or another. It certainly was not our intention to imply otherwise. We appreciate that the way it was written is not completely clear on this point.

**This sentence was re-written in the revised manuscript (page 1, line 11 to 12 in the revised manuscript; page 1, line 11 to 12 in the marked up version).**

*P2. L6-L7. What would be the effect of considering other design situations besides the power production, such as parked conditions? I would suggest adding a short clarification about this.*

Since we have not attempted this in our study, we cannot say for sure. However, it would most likely depend on how sensitive the fatigue damage in these design situations is to changes in the design. Or, specifically, whether or not the fatigue damage in these cases

changes in ways that are proportional to the changes for power production. If the changes are proportional, then the method would still work as reported here. If not, then the results would be weakened, though it is hard to say by how much. We agree that, although the method has been designed specifically to reduce the very large computational effort associated with analyzing the power production DLC, a clarification of this point would be both instructive generally speaking and could also point the way for further studies.

**A paragraph acknowledging and discussing this issue was added to section 4.4 (page 19, line 14 to 21 in the revised manuscript; page 20, line 5 to 12 in the marked up version).**

*P3. L22. Considering normal stresses means that the damage is estimated assuming under uniaxial stress states. How real is this assumption for these type of structures which are normally subjected to multiaxial stress states? What would be the effect of considering multiaxial stress states in the proposed model?*

It is true that these structures are in reality subject to multiaxial stress. However, it has long been standard practice in the industry, and hence also in a lot of research aiming for industrial applications, to consider normal stress only. The main reason for this is that it allows the use of beam models and therefore a much less involved structural analysis when calculating stress and fatigue damage. To properly calculate multiaxial stress means having to use shell elements instead of beam elements in the finite element analysis, which requires more effort both on the modeling side and on the computational side. Additionally, the calculation of fatigue damage from multiaxial stress is much more involved, especially when it comes to the choice of methodology for identification and counting of stress cycles (see e.g. Stephens 2001, Metal fatigue in Engineering, for more on the multiaxial approach). Based on decades of experience from the oil and gas industry (see e.g. Lotsberg 2016, Fatigue design of marine structures) it has been seen that the normal stress approach, while not as accurate, gives reasonable (and often conservative) estimates for the fatigue damage in marine structures. While it would certainly be valuable to understand more about the behavior of the structure under multiaxial stress and how it would affect methods like the one proposed in this paper, especially since the more involved calculations in this case would be an even higher incentive to simplify the load case analysis, this falls outside of the scope of the present work. Without any data to base it on, any notions about how multiaxial stress might affect the effectiveness of the proposed methodology (e.g. more local behavior or larger dependence on directionality) would be highly speculative. However, all that being said, it would be instructive for the reader to understand the possibility of limitations of the method for applications to analysis based on multiaxial stress calculations. We will therefore make it more clear that the presented results are in principle only valid for approaches based on normal stress calculations and that the effectiveness of the method could be reduced if used with multiaxial stress analysis.

**A paragraph discussing this issue and acknowledging the corresponding limitation of the method was added to section 4.1 (page 17, line 11 to 20 in the revised manuscript; page 17, line 32 to page 18, line 6 in the marked up version).**

*P3. L26-L27. Do the authors mean: the maximum value of the total damage among the eight points after evaluating all possible load cases? If so, make a clarification.*

No. This is done per load case. For each load case, the stress is calculated at eight points around the circumference of a given location in the structure (e.g. mudline). The fatigue damage resulting from each stress value is compared and the largest one is selected to represent the fatigue value of this location in the structure, for this specific load case. We can see how the original phrasing of this was a bit unclear.

**The description of this procedure in section 2 was updated to make it more clear (page 3, line 20 to 24 in the revised manuscript; page 3, line 29 to 34 in the marked up version).**

*Regarding Fig. 1-b Does the Normalized fatigue damage correspond to $D_k/D_{tot}$? If so, add clarification in the figure. How was the proportion of total load cases calculated? How Fig. 1-b would look for the different evaluated points along the tower?*

Yes, the partial sums were normalized to the total sum. We agree that making this more explicit also in the figure would be instructive.

**The y-axis label in Fig. 1(b) (top of page 4 in the revised manuscript; top of page 5 in the marked up version) was updated to make this more clear.**

The proportion of total load cases was calculated as the number $k$, the number of load cases used to calculate the k-th partial sum, divided by the total number of load cases considered (in this case 3647). This could also have been more clearly stated in the figure/caption.

**The x-axis label in Fig. 1(b) (top of page 4 in the revised manuscript; top of page 5 in the marked up version) was updated to make this more clear.**

The equivalent curve for the two other points along the tower would be essentially identical and including them in the plot would give little or no additional information. However, this could have been noted in the text so it is more clear that we are not cherry picking the data.

**The text in section 2.1 (page 4, line 10 to 11 in the revised manuscript; page 4, line 18 to 19 in the marked up version) now makes this fact explicit.**

*P4. L11. What does it mean "small" and "intermediate" values of k? How is that scale defined?*

This wording is indeed imprecise. What was meant was the values of k for which the curve in the figure is seen to approach very closely to the asymptote. The language here was perhaps too reliant on a qualitative judgment and should have been quantified more clearly, with reference to the figure.

**The relevant part of this paragraph was re-written to make the statement more quantitative (page 4, line 7 to 9 in the revise manuscript; page 4, line 15 to 18 in the marked up version).**

*P6. L18. How many random seeds were used for each load case in this study? What would be the effect of the number of seeds on the final number of load cases to be evaluated?*

For the sake of testing this method, only one seed per 10 minute time series (with given wind speed, sea state and direction for the incoming waves) was used. To conform with standards, a minimum of six seeds (or equivalently, one 60 minute time series) per load case should have been used. However, the use of additional seeds would only serve to stabilize the fatigue values per load case against random fluctuations. If there would be noticeable effect on the results, it would hence be a beneficial one. On the other hand, this information could very well be of interest to the reader and should be stated in the text.

**A few sentences covering this was added to the relevant paragraph in section 2.4 (page 7, line 13 to 18 in the revised manuscript; page 7, line 10 to 15 in the marked up version).**

*P6. L28-L31. Could the authors elaborate more about how was the scaling process of the element sizes carried out? Were the element sizes scaled only once or several times until the optimal solution was found?*

The elements were scaled just once to obtain each of the models used for testing the method. No further scaling was done. In fact, though each model was meant to "simulate" an optimization process, no actual design optimization was done at any point of this study. Clearly our explanation of the testing setup was not clear enough about what was being done.

**The relevant paragraph in section 2.4 was updated to make the description of the testing methodology more clear (page 7, line 24 to 34 in the revised manuscript; page 8, line 1 to 12 in the marked up version).**

*P7. L3-L6. The statement "From the distribution shown. . ..." is not clear from Fig. 2-a. In this figure, no wind speeds are shown but load cases, which are not clear either. In addition, how can be proved that the load cases with highest normalized fatigue damage are those having the highest probability of occurrence? Is there any reference or way to show this? What does it mean "Normalized fatigue damage"? If you want to show the level of severity, why are you plotting the normalized fatigue damage instead of severity level? I would suggest explaining better this figure both in the figure itself and in the text.*

To your first point, the load cases were grouped according to wind speed, with the smallest wind speed to the left and the largest wind speed to the right (as stated in the figure caption). However, this could have been more clear from simply looking at the figure itself. At least, each wind speed bin could have been marked on the x-axis and the extent of each bin could have been delineated more clearly.

**Figure 3(a) (previously Fig 2(a)) was updated to include more information about the load cases, both on the x-axis and within the plot itself (top of page 8 in the revised manuscript; top of page 9 in the marked up version).**

The rest of your points here follow from an unfortunate error on our part which resulted

from not being consistent with our own terminology. What is being plotted here is really the severity, normalized to the largest value. Hopefully this explains the rest as well. For example, the fact that the load cases with the highest severity for a given windspeed are the ones with the highest probability of occurrence is simply a direct observation that we have made when comparing the index of the peaks in each bin with the scatter diagrams used. Displaying this information explicitly in the plot would be very difficult and it was simply meant as a small observation. However, we should have been consistent with the use of the term severity also in the figure labels and captions.

**The y-axis label and caption of Fig 3(a) were updated to use severity, consistent with the rest of the paper.**

*P8. L10. Regarding the statement "However, this turns out to not be the case.", is this statement for this specific case or in general? If it were for this specific case, what would be the consequences on the proposed model in those cases when the sampling sets are much larger than the number of load cases at each location? If it were in general, how can you prove this statement?*

In a certain sense, we can of course only verify this statement as far as the loading conditions and support structure models used in the study are concerned. We do not believe there is anything particularly special about the setup in a way that would make it simplify this behavior compared to other setups, but cannot prove this in practice. If this behavior breaks down, such that the load cases selected from each location were generally not the same, then the efficiency of the method would reduce by up to a factor 3 (in this case, more generally a factor equaling the number of locations). In order to not make it seem like we were making very general conclusions from these results, we should probably have written something like "our results indicate that this is not the case." A few words about the consequences if this result breaks down for other setups could also be added in the discussion section.

**The relevant sentence in section 3 (page 9, line 16 in the revised manuscript; page 9, line 10 in the marked up version) was changed accordingly.**

**A paragraph discussing this issue in more detail was added to section 4.1 (page 15, line 3 to page 16, line 3 in the revised manuscript; page 17, line 13 to 21 in the marked up version).**

*P8. L14-L16. It would be good to show Fig. 2-b for the three evaluated points. That would provide more veracity to the statement given in this paragraph.*

Figure 2(b) [now 3(b)] does in fact include all three points. The horizontal axis is the number of load cases taken from each point and the vertical axis is the corresponding total size of the sampling set. In other words, selecting the, e.g., 20 most severe load cases from each point results in a total sampling set size of around 25, and so on. Hopefully this does indeed provide the desired veracity to the statement in the text. However, perhaps this could have been made even more clear in the text and/or the figure caption.

**The caption of Fig. 3(b) (previously Fig. 2(b)) (top of page 8 in the revised**

manuscript; top of page 9 in the marked up version) and the relevant parts of the text in section 3 (page 9, line 16 to 17 in the revised manuscript; page 9, line 10 to 11 in the marked up version) were both updated to make this issue more clear.

*P9. L25-L29. Regarding the statement, "We observe that the method seems to consistently over-predict the fatigue damage. . ." What is the consequence of this? Could there be cases in which the results obtained by the method can lead to under-estimated designs (which are not desirable in any structural design)?*

It is indeed possible that the method can in some cases under-predict the fatigue damage of a given design. However, this under-prediction would then only be at the reported error level. From a more practical point of view, there is of course a large difference between an error that leads to under-prediction (unsafe) and an error that leads to over-prediction (safe). Any method like this would have some amount of error, but the difference is that in our case we see that the tendency to over-predict or under-predict the fatigue damage is highly correlated with how the design has been changed compared to the reference design. It would hence be possible for a designer to apply a safety margin in the case where there would be a strong indication that there could be a slight under-prediction of the fatigue damage. We tried to explicate this fact in the discussion section (Page 12, line 3 to 7), but could perhaps have been even more clear in our discussion about these ideas.

**The relevant discussion in section 4.1 was expanded and re-written for clarity (page 13, line 30 to page 14, line 3 in the revised manuscript; page 17, line 2 to 6 in the marked up version).**

*Regarding Figures 3 and 4. If the error can have both negative and positive values, it means that the estimated damage value could be greater than the real damage value. How could be that possible? So, how would you choose the optimal sample set size which makes a balance between the number of loads to evaluate and the final accuracy? Would it be possible to find this value by implementing a simple optimization process? How is the behavior after 180 load cases? It would be good to show more results taking into account that the real number of load cases is larger than 3000. In this way, you could show with more confidence the accuracy of the model.*

The estimated fatigue value is simply a scaling of the fatigue value of the initial design. If the scale factor is too large, then the estimated damage value would be larger than the real damage.

In a certain sense, we have left the choice of "optimal" sample set size to the reader. We could have set a target accuracy and simply increased the sample set size until this was reached. However, such a target value would be entirely application specific. Some might be happy with an error of 10%, others might want an error less than 1%. Furthermore, the exact sample set size is likely highly dependent on the support structure design, turbine model and loading scenario. While we expect based on our results that sample set sizes of 30-50 would give errors of only a few percent or less (except for designs that have been altered to a large extent), giving an explicit number would limit the results to only the

specific case(s) studied.

We terminated our figures at sample set sizes where the error started to flatten out. We also had in mind that for the method to be considered worthwhile, at least when compared to other methods proposed in the cited previous studies, the number of load cases used needed to be small when compared to the total number of load cases (3647). However, we do see the utility of including larger sizes, at least up to reductions of a factor of 10 (approximately 360 load cases), or even 5 (approximately 720 load cases). This would show more clearly the convergence of the results (indicating that including more load cases gives little additional reduction in error). Perhaps such extended curves would also make it easier for the reader to choose their own "optimal" sample set size.

**Two additional figures showing the behavior of the method for larger sets of load cases were added as Fig. 5 (top of page 12 in the revised manuscript; top of page 12 in the marked up version) and Fig. 7 (top of page 15 in the revised manuscript; top of page 15 in the marked up version).**

**Furthermore, two corresponding paragraphs commenting on these figures were added to section 3.1 (page 10, line 15 to 23 in the revised manuscript; page 10, line 28 to page 11, line 2 in the marked up version) and to section 3.2 (page 11, line 3 to 5 in the revised manuscript; page 13, line 4 to 6 in the marked up version) respectively.**

*P9. L6-L7. Regarding the statement, "This in turn makes. . ." What would be a possible solution for this?*

The most straightforward solution to this issue is to make sure that the changes in the eigenfrequency of the structure are within a certain tolerance. Specifically, that the structure does not approach regions of significant dynamic amplification for certain wind speeds (as indicated by the Campbell diagram of the turbine model). If such a situation is reached, one would either have to "restart" the method by doing a new full fatigue analysis or one would have to apply a safety factor to ensure that the increased uncertainty would not lead to unsafe designs. This was discussed in the discussion section (Page 13, lines 29-33 and Page 14, lines 1-2), but could have been more explicit in its connection with specific results.

**The relevant discussion in section 4.4 (page 18, line 32 to page 19, line 9 in the revised manuscript; page 19, line 22 to 33 in the marked up version) was re-written and expanded to make it more clear and to connect it more explicitly with the results.**

*P10. L7-L12. Not sure how pertinent is this discussion for the purpose of this paper.*

This discussion was originally meant to show that the behavior seen for designs MD5, MI5, MD10, MI10 was also seen for designs MR5, MR10 and MRU10. It was further meant to shed some light on why that might be the case, looking into the overall change in mass for these designs. However, during the revision of the manuscript it was discovered that the results shown for the latter designs were misleading in terms of this behavior.

Specifically that it did not hold for all locations in the structure. Hence, these lines needed to be changed in order to reflect the overall results more correctly. Rather than remove this part completely, we decided to shorten and change it, since it still reflects back on behavior observed for the other designs and gives some clues as to what causes this behavior. This change does not have a major impact on the overall results of the study, nor does it have a significant impact on later discussions in the paper, but some changes needed to be made in order to correct this mistake in the original manuscript.

**The relevant sentences (page 10, line 34 to page 11, line 2 in the revised manuscript; page 12, line 6 to page 13, line 3 in the marked up version) were shortened and changed as referenced above.**

**Furthermore, some changes were made to the discussion in section 4.1 (page 13, line 31 to page 14, line 4 in the revised manuscript; page 17, line 2 to 7 in the marked up version) to reflect these differences and a few sentences were added to discuss the corrected results (page 14, line 5 to page 15, line 2 in the revised manuscript; page 17, line 9 to 12 in the marked up version).**

*According to this section 3.3., the level of accuracy of the proposed model could decrease considerably when many points in the structure are analyzed since the sample set size could much higher than the number of load cases at each point (i.e. n>>k). How could this limitation be controlled? This is especially important when the entire structure is analyzed under fatigue.*

Here it seems we have not been clear enough in our explanations. The absolute level of accuracy does increase, it is merely that the relative accuracy does not. There are two reasons for this. One is that as $\epsilon_k$ and $\epsilon_k^{\text{new}}$ become very small for larger values of $k$, the relative accuracy becomes less meaningful (a relative difference of 100% at numerical values of $10^{-8}$ is hardly a significant error in practice). The second, and perhaps more convincing, reason is that while both parts of the fraction in Eq. 8 do tend toward zero, the denominator will tend to zero faster. This does not mean that accuracy is decreasing for higher values of $k$, it merely means that there is little gained for the method in general once $k$ increases past a certain point. The convergence towards the exact answer is very slow, as each additional load case gives very little new information. All in all, this could have been more clearly written and explained in the text.

**Section 3.3 (page 12, line 6 to page 13, line 15 in the revised manuscript; page 13, line 12 to 31 in the marked up version) was re-written and expanded to make the arguments more clear. Additionally, the plots in Fig. 8 (previously Fig. 5) (top of page 16 in the revised manuscript; top of page 16 in the marked up version), as well as the corresponding caption, were changed to display the results in a way that more clearly illustrate what is going on and make the arguments in section 3.3 more evident.**

As for the effect of including more points, we think that having selected three points at large separation in the structure, and having shown from Figure 2(b) [now 3(b)] that the number of additional load cases needed in the sample set is fairly small, that the results obtained would indicate that there should be little additional computational effort or loss

of accuracy if the entire structure is analyzed.

*Regarding Fig. 5, Add the location of the point along the tower at each plot of the figure. How is the error shown in Figure 5 for a greater number of load cases, e.g. 200, 500, 1000?*

The location along the tower of each subplot, though indicated in the main caption, could certainly be included in each sub-plot.

As discussed above, the relative error shown in Figure 5 [now 8] will in fact increase for a larger number of load cases. While some additional information could have been included here, the values of $k$ selected were meant to show the behavior at the values determined most relevant from the previous results. However, to illustrate this point a bit more clearly, we could certainly add a few more rows (for example for the suggested values of $k$) to the figure.

**The already mentioned new version of Fig. 8 (previously Fig. 5) (top of page 16 in the revised manuscript; top of page 16 in the marked up version) also includes the location in each sub-plot and a few more rows for larger values of $k$.**

*Regarding section 4.1, I would suggest analyzing the viability and the limitations of the proposed model in a general point of view instead of focusing only in the evaluated optimization methodologies (i.e. MD5, MD10, etc.). The readers might have other optimization methodologies and it would be useful for them to know when they can implement this method.*

Referring back to our reply regarding the explanation of the testing setup, we would again stress that the different structural models analyzed (MD5, MD10, etc.) do not constitute different optimization methodologies, but rather different fixed states of the structure that would likely be encountered during many types of optimization contained within the relevant scope of the application (mass/weight optimization of the support structure). Of course we cannot claim to have covered everything and we could perhaps have discussed some possible limitations related to this, but overall we think that focusing precisely on the behavior of these test scenarios represent the best way to draw a more general set of conclusions about how the method might behave in different situations. Some further discussion to underline how these models represent (or do not) relevant scenarios that could motivate the reader to use the method for their own purposes will be added.

**A paragraph discussing this issue was added to section 4.2 (page 18, line 1 to 10 in the revised manuscript; page 18, line 23 to 32 in the marked up version).**

*P12. L10 to P13. L3. Elaborate more on these statements, they are not clear as they are now.*

This was meant to also refer back to some of the discussion of the results in the previous section, but it was clearly written either too briefly or not clearly enough.

**The relevant paragraph in section 4.2 (page 17, line 23 to 29 in the revised**

manuscript; page 18, line 9 to 17 in the marked up version) has been re-written and expanded to make the arguments more clear.

*P13. L19-L23. I do see important to consider in future works the uncertainty related to the chosen number of load cases k and, even more, the one related to the final sample set size n.*

This is true, though it is hard to do so without performing a rather large and comprehensive comparison of different support structures, different turbine models and different environmental data.

*It would be good to add a diagram summarizing the proposed model.*

A flow chart or similar that summarizes the steps involved in the method could certainly be added.

**A flow chart of the basic steps in the presented method was added as Fig. 2 (top of page 6 in the revised manuscript; top of page 7 in the marked up version).**

*I did not find any comparison or references to previous works during the discussion.*

We did not make any specific references here, since we already discussed some details of previous work in the introduction and hoped that this would still be on the reader's mind. However, we agree that this makes this subsection a bit hard to read in isolation and that making some more specific references would help make the points more clear.

**References to previous studies were added in section 4.3 (page 18, line 13 to 14 and page 18, line 19 in the revised manuscript; page 19, line 4 and page 19, line 9 in the marked up version.**

*P1. L13. Change "a few" for "some"*

Noted.

**Done (page 1, line 13 in the revised manuscript; page 1, line 13 to 14 in the marked up version).**

*P1. L15-L18. The two first sentences (i.e. "A central practical. . .." and "In order to assess..") could be rewritten in a shorter and clearer way.*

Agreed.

**The sentences have been re-written (page 1, line 16 to 18 in the revised manuscript; page 1, line 16 to 21 in the marked up version).**

*P1. L21. Commission*

Noted.

**Done. Since this was a reference to an entry in the reference list (page 21,**

line 14 in the revised manuscript; page 22, line 16 in the marked up version), the change has been marked there.

*P3. L13-L17. I would suggest deleting this paragraph. This information is not necessary.*

Such a paragraph is standard practice in many cases, but it is not essential. We take this feedback under advisement.

**The paragraph was deleted (page 3, line 16 to 20 in the marked up version).**

*P3. L19. It is not clear what the authors mean in the first sentence. Rewrite it. P5. L2. ". . ..of some new designs of the same structure, with. . .."*

Noted.

**The sentence (page 3, line 14 to 15 in the revised manuscript; page 3, line 22 to 24 in the marked up version) was re-written.**

**The suggestion for a change in the sentence (page 5, line 2 to 3 in the revised manuscript; page 4, line 23 to 24 in the marked up version) was implemented with a slight further modification.**

*P6. L5-L7. Write the last sentence of this paragraph also in equations. That would make the idea clearer.*

Noted.

**Equations and some corresponding updates to the surrounding text were added to section 2.3 (page 6, line 4 to 11 in the revised manuscript; page 6, line 13 to 20 in the marked up version).**

*Regarding Fig. 2. Add a legend defining both the green points and the blue points What is the x-axis scale?*

Noted.

**A legend was added to Fig. 3(a) (previously Fig. 2(a)) (top of page 8 in the revised manuscript; top of page 9 in the marked up version).**

The x-axis has no conventional scale, but is rather a collection of all load case indices (initially not shown because there are 3647 indices). As noted previously, we will make some changes to this figure to make this information more clear.

**Changes covering this aspect of the figure have already been noted above.**

*P7. L3. Change "in the left panel of Fig. 2" for "Fig. 2-a"*

**Done, with a slight modification (page 8, line 5 in the revised manuscript; page 8, line 16 in the marked up version).**

*P8. L3. Make clearer which type of design is refereeing in "For each design,. . .".*

**Done (page 9, line 9 in the revised manuscript; page 9, line 3 in the marked up version).**

*P8. L4. "Specifically, the performance of the method has been quantified. . ."*

**Done (page 9, line 11 in the revised manuscript; page 9, line 5 in the marked up version).**

*Regarding Eq. 7, define the variable in the text.*

**Done (page 9, line 12 in the revised manuscript; page 9, line 5 in the marked up version).**

*P8. L11. Change ". . .in the right panel of Fig. 2." for "Fig. 2-b".*

**Done, with a slight modification (page 9, line 17 in the revised manuscript; page 9, line 11 in the marked up version).**

*P8. L11-L12. "It is reasonably linear, varying between n=7 for k=7 and n=181 for k=150."*

**Done (page 9, line 17 to 18 in the revised manuscript; page 9, line 12 in the marked up version).**

*P9. L18. "The relative errors $\delta$ (?) for various sample sizes n (?) is shown. . .."*

It is indeed $\delta$ and $n$ being referred to.

**Done (page 9, line 24 in the revised manuscript; page 10, line 3 in the marked up version).**

*P9. L18. Are the "designs" in this line related to the "new designs" mentioned in P4. L2.?*

Yes. Again, we will make some changes to the description of the testing setup to make this more explicit.

**Changes related to this have been noted previously.**

*Regarding Figures 3 and 4. Add the name of the models (i.e. MD5, MI5, etc.) at each plot of these figures in order that each plot can be understood itself without the need for the reader to read the caption of the entire figure. Is the "Relative error" at the y-axis referred to $\delta$ from Eq. 7? If so, add $\delta$ in the y-axis, as well as the units.*

Noted.

Yes, it is $\delta$ and this will be added. However, since this is a dimensionless number, there are no units.

**Fig. 4 (top of page 11 in the revised manuscript; top of page 11 in the marked up version) and Fig. 6 (top of page 14 in the revised manuscript; top of page 14 in the marked up version) (previously Fig. 3 and Fig. 4 respectively) have**

been updated to include design names on top and $\delta$ in the y-axis label.

*P9. L3. Define "3P frequency"*

**Done (page 10, line 9 to 10 in the revised manuscript; page 10, line 22 to 23 in the marked up version).**

*P9. L9. "The relative errors $\delta$(?)"*

Yes.

**Done (page 10, line 25 in the revised manuscript; page 11, line 4 in the marked up version).**

*P10. L9. Double "observe"*

**This sentence was deleted as part of a previously mentioned change.**

*P10. L18. Use different notations for the value shown in Eq. 3 and the actual one.*

**Done (page 12, line 6 in the revised manuscript; page 13, line 12 in the marked up version).**

*Regarding Eq. 8, define in the text all variables of this equation.*

**Done (page 13, line 6 to 8 in the revised manuscript; page 13, line 23 to 25 in the marked up version).**

*P12. L10. Change "given that" for "because".*

**This sentence was deleted as part of a previously mentioned change.**

*P13. L3. I would say "With regards to applications to design optimization, this method seems to be very promising"*

**Done, with minor modifications reflecting also some previous changes (page 17, line 29 in the revised manuscript; page 18, line 17 in the marked up version).**

*P13. L10. I would eliminate this title and add section 4.3 to section 4.2*

Since section 4.2 was expanded and section 4.3 was improved according to previous comments, we think these sections work best separately.

*P13. L28-L30. Elaborate more on this idea and change "(e.g.)" for ", e.g.," or ", for example,"*

This idea is in fact continued in the next few sentences of the paragraph, but this could have been written in a more clear way.

**This paragraph was re-written and extended to make the ideas more clear (page 18, line 32 to page 19, line 9 in the revised manuscript; page 19, line 22**

to 33 in the marked up version). The specific change requested was included as part of this.

*The text is full of informal language, like the ones shown below. I would suggest using a more formal language (e.g. In other words, shown, etc.).*

Generally speaking, this is a matter of taste. However, there are some examples here that probably push into informal language in a way that may make the text less understandable for some. We will make the necessary adjustments to avoid this, noted below. Otherwise, the remaining text has been left unchanged except as part of previously mentioned changes.

*Substantial machinery in place (P3. L19.)*

**This was modified as part of a previous change.**

*Effectively speaking (P5. L5.)*

**This was simply deleted (page 4, line 27 in the marked up version).**

*In plain words (P5. L21.)*

**Done (page 5, line 22 in the revised manuscript; page 6, line 3 in the marked up version).**

*That is to say (P7. L7.)*

**Done (page 9, line 4 in the revised manuscript; page 8, line 20 in the marked up version).**

*this turns out (P8. L10.)*

**This was modified as part of a previous change.**

*displayed (P10. L18.)*

**Done (page 12, line 7 in the revised manuscript; page 13, line 13 in the marked up version).**

*Put in another way (P13. L21.)*

**Done (page 18, line 23 in the revised manuscript; page 19, line 13 in the marked up version).**

**Appendix: Marked up version of revised manuscript**

[revised manuscript text omitted]